# UniG: Modelling Unitary 3D Gaussians for View-consistent 3D Reconstruction

## Abstract

In this work, we present UniG, a view-consistent 3D reconstruction and novel view synthesis model that generates a high-fidelity representation of 3D Gaussians from sparse images. Existing 3D Gaussians-based methods usually regress Gaussians per-pixel of each view, create 3D Gaussians per view separately, and merge them through point concatenation. Such a view-independent reconstruction approach often results in a view inconsistency issue, where the predicted positions of the same 3D point from different views may have discrepancies. To address this problem, we develop a DETR (DEtection TRansformer)-like framework, which treats 3D Gaussians as decoder queries and updates their parameters layer by layer by performing multi-view cross-attention (MVDFA) over multiple input images. In this way, multiple views naturally contribute to modeling a unitary representation of 3D Gaussians, thereby making 3D reconstruction more view-consistent. Moreover, as the number of 3D Gaussians used as decoder queries is irrespective of the number of input views, allow an arbitrary number of input images without causing memory explosion. Extensive experiments validate the advantages of our approach, showcasing superior performance over existing methods quantitatively (improving PSNR by 4.2 dB when trained on Objaverse and tested on the GSO benchmark) and qualitatively.

## 1 Introduction

3D object reconstruction and novel view synthesis (NVS) are pivotal in computer vision and graphics, converting 2D images into detailed 3D structures in various applications such as robotics, augmented reality, virtual reality, medical imaging, archaeology, and more. Neural Radiance Fields (NeRF) attempts (Xu et al., 2024; Mildenhall et al., 2020; Wang et al., 2021a; Chen et al., 2021; Yu et al., 2021; Liu et al., 2024a; Xiong et al., 2024) are notable in 3D fields recently. However, their progress is impeded by slow rendering speeds due to the implicit. Recently, as a semi-implicit representation, 3D Gaussian Splatting (3D GS) (Kerbl et al., 2023) has achieved remarkable optimization speed and high-quality novel view rendering performance in representing objects or scenes.

However, many recent methods based on 3D GS techniques encounter the challenge of view inconsistency. This issue arises due to imprecise depth estimations from each views leading to duplicated representations of the same object regions within the 3D reconstructions from different perspectives. For instance, MVGamba (Yi et al., 2024) treats images and 3D Gaussians as sequences in Mamba (Gu & Dao, 2023; Dao & Gu, 2024) which leads to input view order induced view inconsistency. Splatter Image (Szymanowicz et al., 2024) and LGM (Tang et al., 2024a) predict pixel-aligned 3D Gaussians for each input view in the camera space of the corresponding input view. Then these 3D Gaussians are transformed from camera spaces of each view to the world space and naively merged together to obtain the ultimate 3D Gaussians, as depicted in fig. 1(a).

However, such dimension lifting from 2D images to 3D Gaussians in different views are independent and lacks interactions among different views, thus it may result in a single object point being represented by multiple 3D Gaussians at different positions, leading to the aforementioned view-inconsistency issue (Yang et al., 2024; Dong & Wang, 2024).

To address this issue, we propose a **Uni**tary 3D **G**aussians (UniG) representation. Inspired by Deformable DETR (Liu et al., 2023b; Li et al., 2024; Zhang et al., 2023; Liu et al., 2022; Li et al., 2023a) that treats the position and properties of bounding box (Bbox) as queries of the Transformer

| Method | PSNR ↑ | SSIM ↑ | LPIPS ↓ |
|---|---|---|---|
| Splatter Image | 25.6241 | 0.9151 | 0.1517 |
| LGM | 26.2487 | 0.9249 | 0.0541 |
| MV-Gamba* | 26.2500 | 0.8810 | 0.0690 |
| Our Model | **30.4245** | **0.9614** | **0.0422** |

(a). LGM (Tang et al., 2024a)  (b). Ours  (c). Results

**Figure 1:** (a) Previous methods such as LGM (Tang et al., 2024a) directly concatenate Gaussians from different views, leading to view inconsistency. (b) Our method employs a unitary set of 3D Gaussians, projecting them onto each view and integrating information across views for Gaussian updates. (c) Our approach significantly surpasses previous methods in the novel view synthesis task.

decoder, we develop a DETR-like Transformer encoder-decoder framework, which treats 3D Gaussians as decoder queries and updates their parameters layer by layer by performing cross-attention over multiple input views as keys and values. To work over multi-view input, we propose a multi-view deformable attention (MVDFA) operation, where each 3D point fetches related information from multi-view 2D images simultaneously, effectively guaranteeing the consistency. More specifically, MVDFA utilize camera modulation techniques (Karras et al., 2019; Hong et al., 2024) to diversify queries based on views. The queries are linearly transformed to make difference in each view, with the weights and bias trained from camera parameters. Such operation gives each view its corresponding camera pose information. The view-specific queries are then used for performing deformable attention over corresponding images. Although similar to DFA3D (Li et al., 2023a) and BEVFormer (Li et al., 2022) in employing deformable attention in 3D with a point projection strategy, our model prioritizes multi-view distinctions (different qureies in different views) to achieve a more precise 3D representation. Further elaboration is available in section 2.

As the number of 3D Gaussians is usually very large, e.g. over 10,000, the self-attention operation in a deformable Transformer decoder layer will demand a significant memory and computational cost. To improve the efficiency, inspired by (Wang et al., 2021b), we introduce a 3D Spatial Efficient Self-Attention (SESA) approach, leveraging Fast Point Sampling (FPS) (Qi et al., 2017a) to downsize the number of keys and values while preserving the number of queries. Moreover, directly regressing the positions of 3D Gaussians may lead to convergence challenges (see appendix A.4). To address this problem, we utilize a coarse-to-fine framework, where a direct lift from 2D to 3D is employed for every pixel in randomly selected input views at the coarse stage. Then, the 3D Gaussians from this stage serve as the initialization for the deformable Transformer-based refinement network, facilitating meaningful projected positions and aiding in convergence.

In summary, our contributions are as follows:

- We propose UniG, a novel 3D object reconstruction and NVS algorithm which utilizes a unitary set of 3D Gaussians as queries in deformable Transformer. Such an approach allows all input views to contribute to the same 3D representation and effectively addresses the view inconsistency issue and supports arbitrary number of input views.

- We propose to use MVDFA for tackling the multi-view fusion challenge, SESA for minimizing the memory usage in self-attention, and a coarse-to-fine framework for mitigating the convergence issue when directly regressing world coordinates of 3D Gaussians.

- Both quantitative and qualitative experiments are conducted for evaluation. Our proposed method achieves the state-of-the-art performance on the commonly-used benchmark.

## 2  RELATED WORK

**3D reconstruction from images**   Recently, various methods have been explored to reconstruct detailed 3D object from limited viewpoints. (Liu et al., 2024b;c; Tang et al., 2024b; Song et al., 2021a) view the problem as an image-conditioned generation task. Leveraging pretrained generative models like Rombach et al. (2022), they achieve realistic renderings of novel views. However, diffusion models require longer time to generate 3D with multi-step denoising process, thus limiting their applicability in real-time scenarios. Recent methodologies that rely on a single forward pro-

cess for 3D reconstruction, utilizing Neural Radiance Field (NeRF) (Mildenhall et al., 2020) as a robust 3D representation, have demonstrated effective performance in the field of 3D reconstruction. (Yu et al., 2021; Cao et al., 2022; Guo et al., 2022; Lin et al., 2022; Li et al., 2023b; Müller et al., 2022; Liu et al., 2024d; Wei et al., 2024; Tochilkin et al., 2024; Xu et al., 2024; Yu et al., 2021; Wang et al., 2021a; Chen et al., 2021). However, due to the slow rendering speed of NeRF, it is being supplanted by a new, super-fast, semi-implicit representation—3D Gaussian Splatting (3D GS) (Kerbl et al., 2023). Triplane-Gaussian (Zou et al., 2024), Gamba (Shen et al., 2024), and DIG3D (Wu et al., 2024) make promising results on single image 3D reconstruction. Various techniques such as SplatterImage (Szymanowicz et al., 2024), LGM (Tang et al., 2024a), pixelSplat (Charatan et al., 2024), and MVSplat (Chen, Yuedong and Xu, Haofei and Zheng, Chuanxia and Zhuang, Bohan and Pollefeys, Marc and Geiger, Andreas and Cham, Tat-Jen and Cai, Jianfei, 2024) have extended the application of 3D Gaussian Splatting to multi-view scenarios. In these approaches, each input view is processed to estimate 3D Gaussians specific to the view, followed by a simple concatenation of the resulting 3D Gaussian assets from all views. GS-LRM (Zhang et al., 2024) and GRM (grm, 2024) exhibit a model structure similar to LGM, resulting in notable accomplishments through enhanced training processes and consequently more precise depth regression. Nevertheless, these models adhere to the pipeline of predicting 3D Gaussians separately for each view, they demands substantial computational resources, particularly as the number of views grows, the number of Gaussians scales linearly with the number of views. Furthermore, these methods are unable to accommodate an arbitrary number of views as input.

**Deformable Transformer in 3D**   DFA3D (Li et al., 2023a) and BEVFormer (Li et al., 2022) are introduced to address the feature-lifting challenge in 3D detection and autonomous driving tasks. They achieve notable performance enhancements by employing a deformable Transformer to bridge the gap between 2D and 3D. DFA3D initially uses estimated depth to convert 2D feature maps to 3D, sampling around reference points for deformable attention in each view. However, the 3D sampling point design causes all projected 2D points to represent a singular point, neglecting view variations. BEVFormer (Li et al., 2022) regards the Bird's-Eye-View (BEV) features as queries, projecting the feature onto each input view. The Spatial Cross-Attention facilitates the fusion of BEV and image spaces, though challenges persist sampling 4 height values per pillar in the BEV feature for selecting 3D reference points may limit coverage, posing challenges in accurate keypoint selection for the model. When contrasting DFA3D and BEVFormer with our MVDFA, a commonality lies in projecting onto 3D regression targets to extract data from various image perspectives. However, our model diverges by employing camera modulation to differentiate queries across views, enabling more specific information retrieval.

## 3  METHODS

### 3.1  PRELIMINARIES OF 3D GS

3D GS (Kerbl et al., 2023) is a novel rendering method that can be viewed as an extension of point-based rendering methods (Kerbl et al., 2023; Chen & Wang, 2024). Hence, 3D Gaussians can serve as effective 3D representations for efficient differentiable rendering. Each 3D Gaussian ellipsoid can be described by $\mathbf{G} = \{\text{SH}, \boldsymbol{\mu}, \boldsymbol{\sigma}, \mathbf{R}, \mathbf{S}\}$. The color of 3D Gaussians is represented by spherical harmonics ($\text{SH} \in \mathbb{R}^{12}$) while the geometry is described by the center positions $\boldsymbol{\mu} \in \mathbb{R}^3$, shapes (covariance matrix $\boldsymbol{\Sigma}$), and opacity ($\boldsymbol{\sigma} \in \mathbb{R}$) of ellipsoids (Zwicker et al., 2001; Kerbl et al., 2023). Especially, the covariance matrix can be optimized through a combination of rotation and scaling for each ellipsoid as $\boldsymbol{\Sigma} = \mathbf{R}\mathbf{S}\mathbf{S}^T\mathbf{R}^T$, where $\mathbf{R} \in \mathbb{R}^4$ (represented by quaternion) represents the rotation and $\mathbf{S} \in \mathbb{R}^3$ contains the scales in three directions.

### 3.2  OUR METHOD

**Overall framework**   As illustrated in fig. 2, our model follows an encoder-decoder framework in a coarse-to-fine manner. We employ unitary 3D Gaussian representation, which define a unitary set of 3D Gaussians in the world space no matter how many input views are given. During the coarse stage, one or more images are randomly selected as input for a simple encoder-only model to directly predict 3D Gaussians, supervised by a RGB loss. Subsequently, in the refinement network, all input

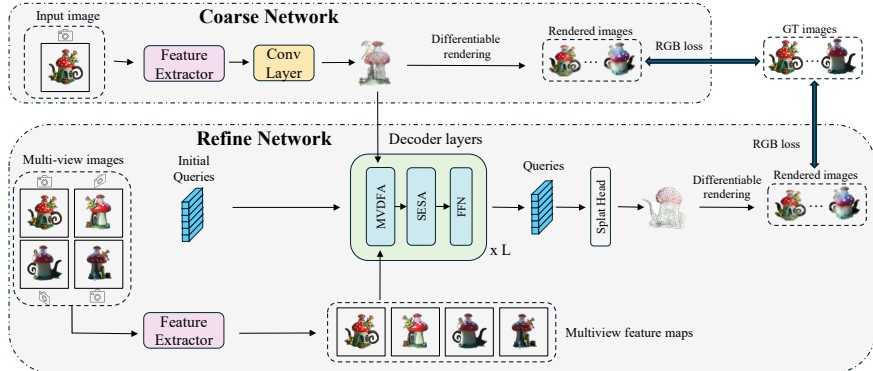

**Figure 2:** Overall Framework: In the coarse stage, 3D Gaussians are produced for each pixel of the sampled random views from the input data. In the refinement stage, 3D Gaussians from the coarse stage serves as the initialization for the refinement network. Multi-view features extracted by the feature extractor serves as keys and values of decoder. Queries are updated by the decoder layer with image features and the positions of the centers of 3D Gaussians. The final 3D Gaussian representation is regressed from the queries. MVDFA: multi-view deformable attention in section 3.2.2. SESA: spatial efficient self-attention in section 3.2.3. FFN: feed-forward network.

images undergo processing through an image encoder and a cross-view attention module to extract multi-view image features (section 3.2.1).

Each 3D Gaussian is then projected onto each view to query relevant features and update their respective parameters by query refinement decoder with multi-view deformable attention (MVDFA)(section 3.2.2). Spatially efficient self-attention is utilized to reduce computational and memory costs, enabling the utilization of more 3D Gaussians for object reconstruction (section 3.2.3). Moreover, the coarse-to-fine design aims to ensure that the initial positions of the center of 3D Gaussians are not too distant from the ground truth or outside the field of view to gurantee the training convergence (section 3.2.4). The training objective is detailed in section 3.2.5.

### 3.2.1 FEATURE EXTRACTOR

To extract image features from multi-view input, we utilize UNet (Ronneberger et al., 2015; Song et al., 2021b), a widely employed feature extractor in 3D reconstruction tasks, as demonstrated in Tang et al. (2024a); Szymanowicz et al. (2024). To enhance the network's understanding of the complete 3D object, multi-view cross-attention is employed to transfer information among views right after the UNet block, activated when the number of input views exceeds one. In this configuration, each input view acts as queries, while the concatenation (post-flattening) of the remaining views serves as keys and values. To efficiently enable cross-attention across all views, we employ shifted-window attention, as introduced in the Swin Transformer (Liu et al., 2021). This mechanism reduces interactions by focusing on tokens within a local window, effectively reducing memory usage for large input sequences. By processing tokens within a fixed window, shifted-window attention effectively lowers the computational complexity, thereby enhancing the overall efficiency.

### 3.2.2 VIEW-AWARE QUERY REFINEMENT DECODER

**Decoder structure** In the decoder module, we employ a fixed number of queries $\mathbf{Q} \in \mathbb{R}^{N \times C}$ with $N$ and $C$ denote the number of Gaussians and the hidden dimension to model 3D Gaussians by associating queries with 3D Gaussian ellipsoid parameters $G$, including the center $\boldsymbol{\mu}$, opacity $\boldsymbol{\sigma}$, rotation $\mathbf{R}$, scaling $\mathbf{S}$, and Spherical Harmonics $\mathbf{SH}$. As depicted in fig. 2, the queries navigate through multiple decoder layers, each including a multi-view deformable attention (MVDFA) (section 3.2.2) mechanism to leverage image features, a spatial efficient self-attention (SESA) (section 3.2.3) layer for inter-Gaussian interactions, and a feed-forward network (FFN). The functionality of a decoder layer can be summarized by eq. (1), where $\mathbf{F}$ represents image features from different views and $\mathbf{P}^l$

signifies reference points in the $l$-th layer.

$$\mathbf{Q}^{l+1} = \text{FFN}(\text{SESA}(\text{MVDFA}(\mathbf{Q}^l, \mathbf{P}^l, \mathbf{F}))) \tag{1}$$

Finally, queries are processed through a splatter head $\mathcal{S}$ to compute $\Delta\mathbf{G} = \mathcal{S}(\mathbf{Q})$ for updating the 3D Gaussian parameters: $\mathbf{G}' = \mathbf{G} + \Delta\mathbf{G}$ (except for rotation, which is updated by multiplication). Here, all views contribute to unitary 3D Gaussians, emphasizing the most relevant features. This strategy effectively alleviates the view inconsistency issue and is computationally more efficient.

**Multi-view deformable attention (MVDFA)**  The goal of MVDFA is to enhance the unified queries and Gaussian representations by integrating multi-view image features. Following original design of DFA, trainable sampling points are employed on the image features to sample the most relevant image features as values (Carion et al., 2020; Zhu et al., 2021). The remaining problem lies in determining the sampling points and attention scores. Specifically, we project the 3D queries onto each image view and adjust it using camera modulation to account for view discrepancies. The sampling offsets and attention scores are subsequently obtained from the view-specific queries.

By leveraging the center $\boldsymbol{\mu}$ of each 3D Gaussian from the previous layer, along with the corresponding camera poses $\pi_i$ and intrinsic parameters $K_i$ for the $i$-th image, we can compute UV coordinates $\mathbf{P}_i$ by projecting the center coordinates of each 3D Gaussian onto the image plane of the $i$-th input image using the pinhole camera model (Forsyth & Ponce, 2003; Hartley & Zisserman, 2003): $\mathbf{P}_i = K_i\pi_i\boldsymbol{\mu}$. In this context, both matrices $K_i$ and $\pi_i$ are expressed in homogeneous form. These UV coordinates in $\mathbf{P}_i$ then function as the reference points for 2D deformable attention.

As depicted fig. 3 (a), in 3D, we have a set of queries associated with 3D Gaussian paremeters $\mathbf{G}$ while the queries for each image planes should to be adjusted to suit each view individually. To tackle this issue, we start by using camera modulation with the adaptive layer norm (adaLN) (Hong et al., 2024; Karras et al., 2019; 2020; Viazovetskyi et al., 2020) to generate view-specific queries. More information on this modulation is provided in fig. 3(b). Subsequently, a linear layer to predicts the sampling offsets $\Delta\mathbf{s}$ for retriving images features as values and another linear layer to predicts the attention scores $\boldsymbol{\alpha}$ of the sampling points $\mathbf{s}$. Following (Zhang et al., 2023; Zhu et al., 2021), we compute attention scores directly from queries, omitting keys to streamline calculations. Then, we apply the grid sampling algorithm with bilinear interpolation to extract image features at these sampling points, which act as the values $\mathbf{v}$ for cross attention.

Finally, for each input view, we compute the updated queries for each view using the attention scores $\boldsymbol{\alpha}$ and sampled values $\mathbf{v}$. The ultimate unitary queries are then computed as a weighted sum of individual view queries, with the weights calculated using an linear layer on the view-specific queries. Detailed pseudo code for our multi-view deformable cross-attention is available in fig. 3(b).

### 3.2.3 SPATIAL EFFICIENT SELF-ATTENTION (SESA)

Our multi-view deformable cross-attention mechanism demonstrates a superior efficiency in terms of computational cost and memory usage. However, self-attention is computationally expensive, especially with numerous 3D Gaussians. Updating each Gaussian with information from all others may not always be essential, as neighboring Gaussians often contain similar information. To tackle this problem, drawing inspiration from Wang et al. (2021b), we introduce a method to reduce the number of keys and values while keeping the number of queries unchanged during self-attention. This selective update strategy enables each query to be updated with a subset of related queries, effectively enhancing the information exchange efficiency. To ensure crucial information flow, we leverage the Fast Point Sampling (FPS) algorithm from point cloud methodologies (Qi et al., 2017a;b). By utilizing Gaussian centers $\boldsymbol{\mu}$ to identify distant points for querying, we optimize memory usage while guaranteeing essential information sharing among Gaussians. Additional details are in appendix A.1.

### 3.2.4 COARSE-TO-FINE MODEL

**Locating Gaussian centers in the world space**  In Szymanowicz et al. (2024), the Gaussian centers are located in each input view's camera space, i.e. $\boldsymbol{\mu}_{\text{cam}} = [x_{\text{cam}}, y_{\text{cam}}, z_{\text{cam}}] = [u_1 d + \Delta_x, u_2 d + \Delta_y, d + \Delta_z]$, where the center coordinates $x_{\text{cam}}, y_{\text{cam}}, z_{\text{cam}}$ are parameterized by the depth $d$ and offset values $(\Delta_x, \Delta_y, \Delta_z)$. The depth $d$ represents the length of a ray originating from the camera

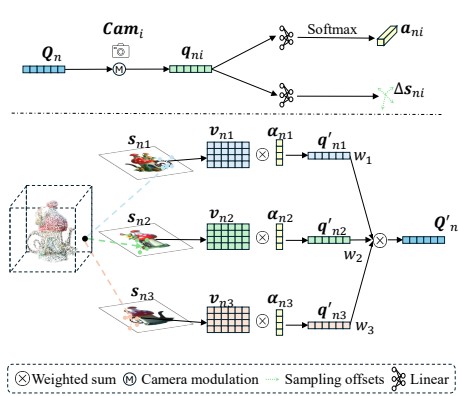

**(a)** MVDFA on the $n$-th 3D Gaussian

```
1  def MVDFA(F, K, π, Q, μ):
2      # Prepare camera embedding
3      camera=concat(K, π).flatten() #[B, I, 16]
4      cam_embed=MLP(camera) #[B, I, C]
5      # Modulate query by cameras, [B, I, N, C]
6      shift, scale=MLP(cam_embed).chunk(2)
7      q=LayerNorm(Q)*(1+scale)+shift #[B, I, N, C]
8      α=softmax(Linear(q)) #Attn score [B, I, N, Ns]
9      # Sampling points
10     Δs=Linear(q) #[B, I, N, Ns, 2]
11     P=pinhole_proj(camera, μ) #[B, I, N, 1, 2]
12     s=P+Δs #[B, I, N, Ns, 2]
13     # Weighted sum of view-specific queries
14     V=Linear(F) # [B, I, H*W, C]
15     v=grid_sample(V, s) #[B, I, Ns, C]
16     q'=(α · v).sum(-1) #[B, I, N, C]
17     w=sigmoid(Linear(q')) #[B, I, C]
18     Q'=(w · q').sum(-2) #[B, N, C]
19     return Q' #[B, N, C]
```

**(b)** Pseudo code

**Figure 3:** MVDFA: $\mathbf{Q}_n$ denotes the $n$-th unitary queries while $\mathbf{q}_{ni}$ denotes the $n$-th query on the $i$-th view modulated by the $i$-th camera $\mathbf{Cam}_i$. Linear layers are used on $\mathbf{q}_{ni}$ to compute the sampling offsets $\Delta\mathbf{s}_{ni}$ and attention score $\boldsymbol{\alpha}_{ni}$. The $n$-th 3D Gaussian is projected onto images, and surrounding sampling points $\mathbf{s}_{ni} = \mathbf{P}_{ni} + \Delta\mathbf{s}_{ni}$ are sampled using offsets $\Delta\mathbf{s}_{ni}$. Values $\mathbf{v}_{ni}$ are image features sampled at $\mathbf{s}_{ni}$. The final query is calculated by the weighted sum of updated view-specific queries $\mathbf{q}'_{ni}$, where $w_i$ is the weight calculated by a linear layer on $\mathbf{q}'_{ni}$. $B$ is batch size, $I$ is the number of views, $C$ is the hidden dimension, $N$ is the number of Gaussians, *pinhole_proj* is the projection from 3D to 2D with the pinhole model. $\mathbf{F}$ is the image feature with height $H$ and weight $W$. $K$ and $\pi$ are camera intrinsics and extrinsics, respectively.

center. $u_1, u_2$ are the UV coordinates of the ray passing through the corresponding input image. This design represents each point with multiple Gaussians, potentially introducing view inconsistency due to concatenation issues at various points caused by depth inaccuracies and tend to shortcut input views (Wu et al., 2024). In our framework, we define unitary Gaussians in the world space, project their centers to each input view for feature retrieval, as depicted in fig. 3(a). The centers of Gaussians can be written as $\boldsymbol{\mu}_{\text{world}} = [x_{\text{world}}, y_{\text{world}}, z_{\text{world}}]$. However, during the initial training phases, discrepancies between the 3D Gaussian centers and ground truth often result in imprecise selection of image features at sampling points, presenting challenges for model convergence.

We employ a relative coordinate system, where the camera poses for all views are known. The initial input view is established as the world coordinates (with the camera pose represented by the identity matrix), and subsequently, all other views are transformed to align with these coordinates. This approach allows us to represent all 3D data within this consistent relative coordinate system.

**Coarse-to-fine**  To address this issue, we utilize a coarse network that directly regress 3D Gaussian parameters with one or more randomly selected input images as input. The role of this network is to provide a coarse initialization of 3D Gaussians for the subsequent refinement network. We use the UNet architecture as the feature extractor to train the coarse network. Subsequently, we use this trained parameters to initialize the refinement stage and independently train the refinement network.

### 3.2.5 TRAINING OBJECTIVE

Building upon prior 3D Gaussian-based reconstruction approaches, we leverage the differentiable rendering implementation by Kerbl et al. (2023) to generate RGB images from the 3D Gaussians produced by our model. For each object, we render 4 input views and 8 additional views (12 views in total) for supervision. Furthermore, aligning with the methodologies ((Hong et al., 2024; Tang et al., 2024a)), we employ a RGB loss in eq. (2), which consists of both a mean square error loss $\mathcal{L}_{\text{MSE}}$ and a VGG-based LPIPS (Learned Image Patch Similarity) loss (Zhang et al., 2018a) $\mathcal{L}_{\text{LPIPS}}$ to guide the rendered views. Here $I_{pd}$ represents the rendered views supervised by the ground truth images $I_{gt}$.

$$\mathcal{L} = \mathcal{L}_{\text{MSE}}(I_{pd}, I_{gt}) + \lambda\mathcal{L}_{\text{LPIPS}}(I_{pd}, I_{gt}) \tag{2}$$

## 4 EXPERIMENTS

This section delves into experiment details and outcomes. In section 4.1, we give dataset specifics, evaluation metrics, and implementation details. section 4.2 delves into quantitative and qualitative results for sparse view novel view synthesis, along with the visualization of 3D Gaussian centers as a point cloud. section 4.3 provides a comparative analysis of processing speeds and memory costs. section 4.4 presents an ablation study. Lastly, The versatility of our model extends to tasks such as image-to-3D and text-to-3D generation using a diffusion model, detailed in section 4.5.

### 4.1 DATASET AND EXPERIMENT SETTINGS

**Dataset** We utilized a refined subset of the Objaverse LVIS dataset (Deitke et al., 2023) for training and validation. The training dataset comprised two sets of rendered images: one set featured 12 random camera poses, while the other included input rendering images captured from fixed viewpoints (front, back, left, right). Supervision was provided from 32 random views spanning elevations between -30 to 30 degrees. The resolution of the rendered images was downscaled to $128 \times 128$.

To evaluate our model, we conducted tests on the Google Scanned Objects (GSO) benchmark. Two test sets were utilized: one with fixed-view inputs (e.g., front, left, back, right) at 0 degrees elevation, tested on 32 random views with elevations ranging from 0 to 30 degrees, and the other includes 25 random views with corresponding camera poses. Importantly, there are no constraints on the elevation of the rendered views. We refer to these test sets as GSO-random and GSO-fixed in our subsequent analysis. More details for dataset can be found in appendix A.2.

**Evaluation metric** We compute the peak signal-to-noise ratio (PSNR), structural similarity index (SSIM) (Wang et al., 2004), and perceptual distance (LPIPS) (Zhang et al., 2018b) between the rendered images and the ground truth. Additionally, we offer visual representations of both the rendered images and the 3D Gaussian centers as a point cloud. More details are provided in appendix A.2.

### 4.2 COMPARISON WITH STATE-OF-THE-ART METHODS

We provide the comparison with the state-of-the-art methods in this section.

#### 4.2.1 FIXED VIEW INPUT

Table 1: Quantitative results for inputting 4 views on GSO-fixed dataset. *The results of MV-Gamba are cited from the paper as they do not provide code or a test set.

| Method | PSNR ↑ | SSIM ↑ | LPIPS ↓ |
|---|---|---|---|
| Splatter Image (Szymanowicz et al., 2024) | 25.6241 | 0.9151 | 0.1517 |
| LGM (Small) (Tang et al., 2024a) | 17.4810 | 0.7829 | 0.2180 |
| LGM (Large) (Tang et al., 2024a) | 26.2487 | 0.9249 | 0.0541 |
| InstantMesh (Xu et al., 2024) | 23.0177 | 0.8893 | 0.0886 |
| MV-Gamba* (Yi et al., 2024) | 26.2500 | 0.8810 | 0.0690 |
| Our Model | **30.4245** | **0.9614** | **0.0422** |

We evaluated recent multi-view reconstruction models using 4 views as input. Splatter Image (Szymanowicz et al., 2024) were trained with their native data loaders, adjusting inputs to 4 views and supervision to 12. LGM and InstantMesh were evaluated using the provided checkpoints, with "Small" indicating models tailored to 128 resolution and "Large" to 256 resolution. All models were assessed assuming the same number of training views.

table 1 showcases the performance of these methods in novel view synthesis using 4 fixed views (front, back, right, left) on the GSO-fixed dataset. Our model surpassed previous approaches in PSNR, SSIM, and LPIPS for novel view synthesis, with a significant improvement of approximately 4.2 dB in PSNR. Additional results for 6 and 8 view inputs are available in appendix A.3.2.

We present visualization results for novel view synthesis in fig. 4 and 3D Gaussian centers represented as point clouds in appendix A.3.1. In our experiments, with resolution of 128, the LGM model corresponds to the small version. Observations in the figures reveal view inconsistency in LGM and a lack of details in InstantMesh, whereas our model maintains both details and view consistency. Further visualizations at a resolution of 256 are accessible in appendix A.3.1.

#### 4.2.2 RANDOM VIEW INPUT

Table 2: Quantitative results for inputting 4 views on GSO-random dataset.

| Method | PSNR ↑ | SSIM ↑ | LPIPS ↓ |
|---|---|---|---|
| Splatter Image | 25.7660 | 0.8932 | 0.2575 |
| LGM | 15.1113 | 0.8440 | 0.1592 |
| InstantMesh | 17.3073 | 0.8525 | 0.1376 |
| Our Model | **26.3020** | **0.9255** | **0.0836** |

Previous methods (LGM and InstantMesh) usually rely on fixed views as input, as they align well with views generated from diffusion models like ImageDream (Wang & Shi, 2023). In real-world scenarios, users are more inclined to provide random views as input. table 2 displays the results when utilizing random 4 views as input on the GSO-random dataset. Notably, there is a performance drop observed in LGM and InstantMesh with random input views. appendix A.3.1 provides the visualization results. For Splatter Image, although the PSNR does not reduced much, its SSIM and LPIPS reduced significantly. We provide more visualization in appendix A.3.1 fig. 14.

#### 4.2.3 INFERENCE ON ARBITRARY NUMBER OF VIEWS

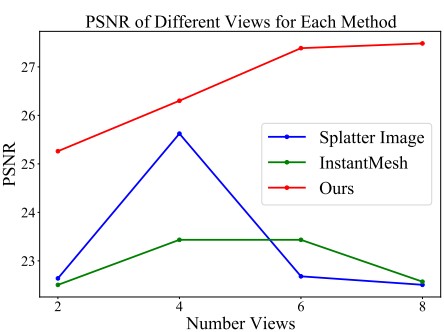

**Figure 5:** Quantitative results with random number of views as input. The model is trained with 4 random input views and tested with variate number of views.

Training costs for 3D methods are considerable, often requiring 32 NVIDIA A100 (80G) GPUs over multiple days. Additionally, memory costs for previous methods increase linearly with the number of views, presenting challenges for training models with varying input views. Therefore, a model supporting inference with any number of inputs while being trained on a fixed set, such as 4 views, would provide significant advantages.

Our model retains unitary 3D Gaussians in world coordinates, treating views as complementary sources without compromising overall 3D integrity. This enables adaptability to variable view counts during inference, despite training on a fixed number of views. fig. 5 showcases the results of training the model with 4 random views and testing it with different number of views. More views results are in appendix A.3.2 fig. 17.

While other methods demonstrate satisfactory performance with 4 views during inference, their effectiveness diminishes as the view count deviates from 4. In contrast, our model excels as the number of views increases. It is important to highlight that LGM is not part of this comparison due to its incapacity to handle variations in the number of views between the training and testing phases.

#### 4.3 INFERENCE TIME AND MEMORY COST

Table 3: Inference time comparison. 3D: forward time, render: rendering time, inference: time of one forward and 32 rendering. Unit in seconds.

| Method | 3D ↓ | Render ↓ | Inference ↓ |
|---|---|---|---|
| DreamGaussian | 118.3245 | 0.0038 | 118.4461 |
| InstantMesh | **0.6049** | 0.6206 | 20.4641 |
| LGM | 1.6263 | 0.0090 | 1.9143 |
| Our Model | 0.6939 | **0.0019** | **0.7538** |

We performed inference time tests across different model types, including a diffusion-based method (DreamGaussian (Tang et al., 2024b)), a NeRF-based model (InstantMesh (Xu et al., 2024)), a previous Gaussian-based model (LGM (Tang et al., 2024a)), and our model, as detailed in table 3. Our model maintains a reduced number of Gaussians and achieves the fastest rendering speed.

In contrast to previous methods that compute 3D Gaussians per pixel per input view, our model retains a single 3D Gaussian irrespective

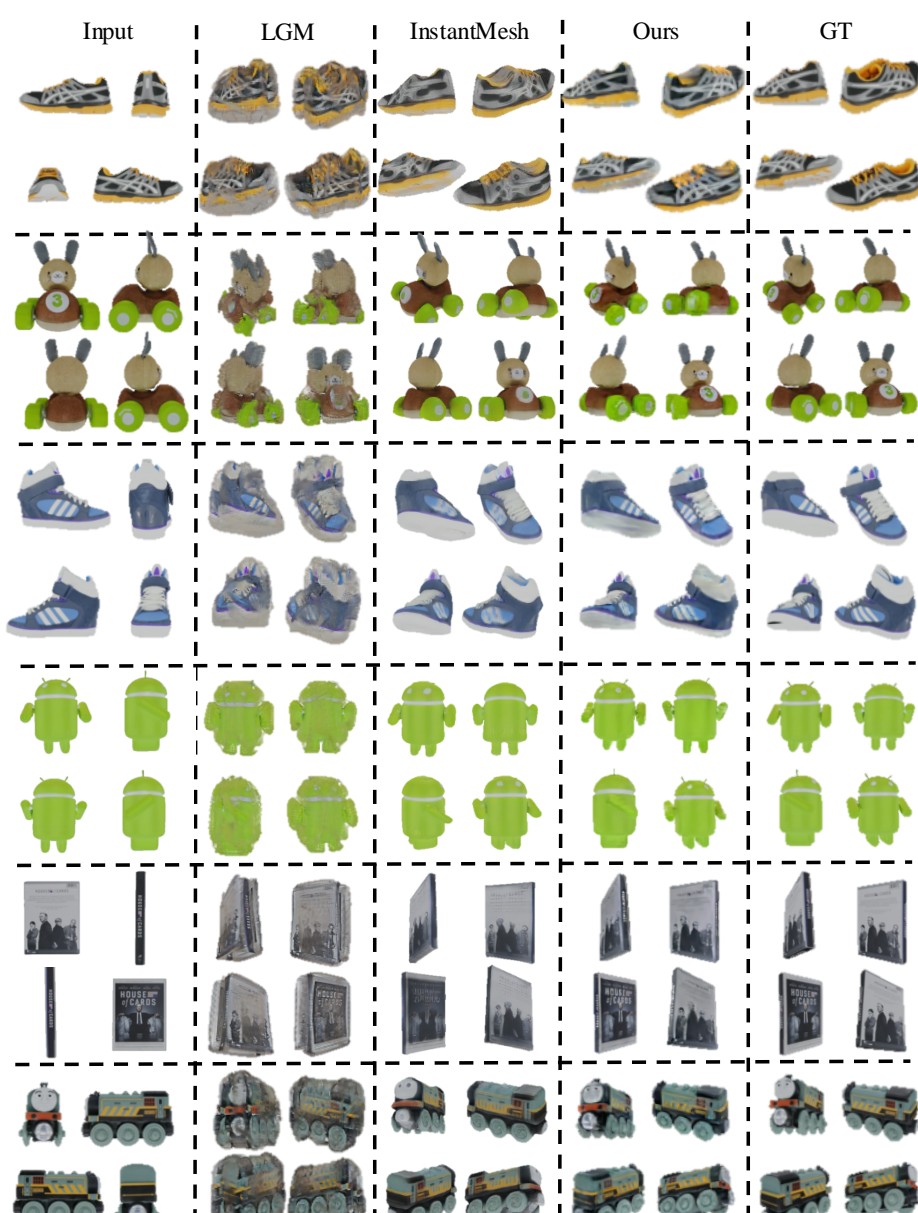

**Figure 4:** Novel views on GSO-fixed dataset for inputting 4 views with resolution 128.

of the number of views. While conventional methods exhibit linear memory expansion with additional views or higher image resolutions, our approach sustains a consistent memory overhead or experiences slight increments due to the marginally higher cost of the image feature extractor. This design theoretically enables our model to accommodate more input views and higher resolutions for enhanced outcomes, potentially circumventing the out-of-memory limitations encountered by other methods.

## 4.4 ABLATION STUDIES

table 4 illustrates an ablation study that evaluates different components of the model architecture. All the experiments are evaluated on the Objaverse validation dataset. Removing the coarse stage and initializing randomly (without any constraint) results in the lowest performance. This problem arises from utilizing image features around the projected 3D Gaussian center within each image view, po-

tentially causing zero features and steep gradients when projections extend beyond the image plane. When the coarse stage is randomly initialized within the cone of vision (CoV), performance improves. To provide a more meaningful initialization, we incorporate a coarse stage to acquire the approximate Gaussian locations, followed by a refinement stage. This refined initialization empowers our model to achieve superior performance. Moreover, removing cross-view attention leads to a moderate decrease in performance compared to the full model. Using only the coarse stage (UNet-based) slightly underperforms the full model. Furthermore, removing the camera modulation on queries or use 3D sampling points instead of sampling on each view adversely impacts the results, underscoring the critical significance of this view-specific design. The full model achieves the best performance across all metrics, indicating that each component contributes positively to the overall model effectiveness. Additionally, we offer details on hyperparameter selections in appendix A.4.

Table 4: Ablation study on model design.

| Method | PSNR ↑ | SSIM ↑ | LPIPS ↓ |
|---|---|---|---|
| w/o coarse (ran. init.) | 12.1213 | 0.6531 | 0.6224 |
| w/o coarse (ran. init. in CoV) | 22.6740 | 0.8711 | 0.2383 |
| w/o cross view attention | 25.3923 | 0.9013 | 0.1007 |
| coarse stage only (UNet) | 25.6033 | 0.9107 | 0.0930 |
| w/o camera modulation | 26.1328 | 0.9201 | 0.0883 |
| 3D sampling points | 25.8392 | 0.9117 | 0.0945 |
| Full model | **26.5334** | **0.9344** | **0.0667** |

Moreover, we add the ablation study on the number of views or different views input in the coarse stage in appendix A.4 table 9. We also give more view inconsistency visualization problem by visualize center of Gaussians from each view in different colors, as shown in fig. 7. Furthermore, removing the background use masks for Splatter Image and LGM may slightly improve the performance (fig. 16, table 6)

### 4.5 APPLICATIONS IN 3D GENERATION

**Image-to-3D** conversion represents a fundamental application in 3D generation. Following the methodology of LGM and InstantMesh (Tang et al., 2024a; Xu et al., 2024), we initially leverage a multi-view diffusion model, ImageDream (Wang & Shi, 2023), to generate four predetermined views. Subsequently, our model is utilized for 3D Gaussian reconstruction. A comparative analysis with LGM and InstantMesh is detailed in appendix A.3.2. We also showcase the quality results of our model on both the GSO dataset and in-the-wild images in appendix A.3.1.

Our model can also do the **Text-to-3D** task. To evaluate quality, we utilize MVDream (Shi et al., 2024) to generate a single image from a text prompt. Subsequently, a diffusion model is employed to produce multi-view images, which are then processed by our model to derive a 3D representation. A qualitative comparison of the text-to-3D generation is presented in appendix A.3.1.

## 5 CONCLUSION AND LIMITATION

In this paper, we have introduced a novel sparse view 3D reconstruction and novel view synthesis method. Initially, a fixed number of 3D Gaussians with predefined properties are initialized, and each Gaussian ellipsoid is projected onto input image features extracted by a feature extractor. We propose the MVDFA block to integrate image features surrounding the projected 3D Gaussians from each view to refine the 3D Gaussians, employing a coarse-to-fine strategy to ensure robust model convergence. Additionally, we develop a spatially efficient self-attention mechanism to minimize computational costs, tackling view inconsistency and computational inefficiency. Our model accommodates an arbitrary number of views as input and showcases its effectiveness through quantitative and qualitative experiments compared to state-of-the-art methods trained on Objaverse and tested on the GSO dataset. Furthermore, with the aid of an off-the-shelf diffusion model, our model undertakes generation tasks such as image-to-3D and text-to-3D conversions. We present an ablation study elucidating the significance of each model component. While our model signifies a notable advancement in sparse view 3D reconstruction, there are inherent **limitations**. Presently, user-provided camera parameters, both camera poses and intrisics, are necessary for projecting 3D Gaussians onto images, presenting potential challenges in 3D reconstruction. Addressing this issue stands as a focal point for future research.

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

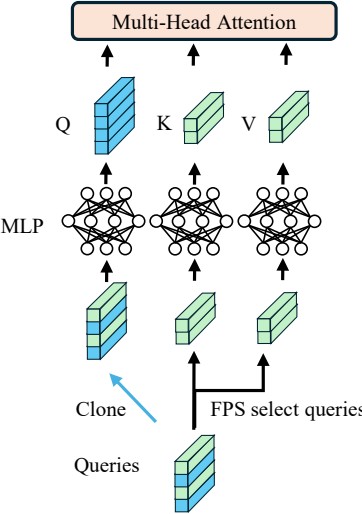

**Figure 6:** Spatially Efficient Self-Attention: While employing all queries as query in the self-attention mechanism, we leverage Farthest Point Sampling (FPS) to downsample certain 3D Gaussians. This process enables the extraction of their corresponding queries as keys and values within the self-attention operation.

# A APPENDIX

## A.1 SPATIAL EFFICIENT SELF ATTENTION (SESA)

While our 3D-aware deformable attention mechanism is notably efficient, the computational cost and memory occupation mainly arises in the self-attention component, particularly when dealing with a large number of 3D Gaussians. However, updating each 3D Gaussian with information from all others is not always necessary because those neighbouring 3D Gaussians usually carry similar information.

To mitigate this issue, as depicted in fig. 6 and drawing inspiration from Wang et al. (2021b), we introduce a technique aimed at reducing the size of the key and value components while maintaining the query component unaltered within the self-attention process. The core concept behind this approach is that while each 3D Gaussian requires updating, not every other 3D Gaussian needs to contribute to this update. We achieve this by selectively updating each query solely with a subset of corresponding queries linked to other 3D Gaussians.

To retain crucial information flow, we leverage the Fast Point Sampling (FPS) algorithm commonly used in point cloud methodologies like PointNet (Qi et al., 2017a) and PointNet++ (Qi et al., 2017b). Specifically, we employ the Gaussian centers $\mu$ to identify the most distantly located points and use these points to index the queries. By implementing this strategy, we significantly reduce the model's overall memory footprint while preserving essential information exchange among the Gaussians.

## A.2 IMPLEMENTATION DETAILS

**Dataset** We utilized a refined subset of the Objaverse LVIS dataset (Deitke et al., 2023) for both training and validating our model. This subset was curated to exclude low-quality models, resulting in a dataset containing 36,044 high-quality objects. This open-category dataset encompasses a diverse range of objects commonly encountered in everyday scenarios. For training, we leveraged rendered images provided by zero-1-to-3 (Liu et al., 2023a) for the random input setting. Each object in the dataset is associated with approximately 12 random views, accompanied by their respective camera poses. We partitioned 99% of the objects for training purposes, reserving the remaining 1% for validation. During training, we randomly selected a subset of views as input while using all 12 views for supervision. Each rendered image has a resolution of $512 \times 512$, which we downscaled to

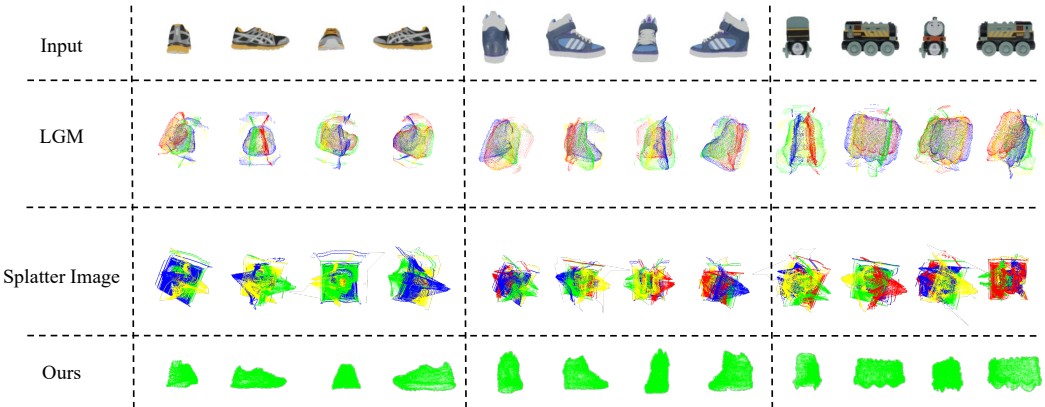

**Figure 7:** Point clouds of the center of Gaussians from each view. The Gaussians from different views are in different colors.

$128 \times 128$. For the fixed view setting, we render the images with fixed views as input and 32 more random views with elevation in $(-30, 30)$ degrees for supervision.

To evaluate our model's performance in open-category settings, we conducted tests on the Google Scanned Objects (GSO) benchmark (Downs et al., 2022). The GSO dataset comprises 1,030 3D objects categorized into 17 classes. For this evaluation, we utilized rendered images sourced from Free3D (Zheng & Vedaldi, 2024), which consist of 25 random views along with their corresponding camera poses. Notably, there are no restrictions on the elevation of the rendered views. We utilized the initial views as inputs and the remaining views for assessing our novel view synthesis task. Additionally, we observed that LGM (Tang et al., 2024a) only support fixed-view inputs (e.g., front, left, back, and right). To address this, we evaluated a new rendered GSO dataset at 0 degrees elevation, testing it on 32 random views with elevations ranging from 0 to 30 degrees. To distinguish between the two test sets, we refer to them as GSO-random and GSO-fixed respectively in the following analysis.

**Experiment setting** We train our model on the setting of 4 views, each time we randomly select 4 views as input and all the views for supervision. In the coarse stage, we train the model with less views (i.e. 2 views) with resolution $128 \times 128$ and generate 16384 3D Gaussians as initialization of the fine stage. In the fine stage, We use 19600 3D Gaussians to represent the 3D object. For the 3D Gaussians from the coarse stage, we use the mask to remove the background points and padding the number of 3D Gaussians to 19600 by copying some of the remaining 3D Gaussians. The selected 3D Gaussians are then utilized to project queries onto image plane in the refine stage. In each deformable attention layer, we utilize 4 sampling points for each projected 3D Gaussian reference point to sample values on the image.

We use 4 decoder layers and the hidden dimension is 256. Moreover, when training the fine stage, we finetune both the coarse stage and the encoder. We use a mixed-precision training (Narang et al., 2018) with BF16 data type. We train our model with Adam (Kingma & Ba, 2015) optimizer and the learning rate is 0.0001. We take 300K iteration with batch size 4. For the coarse stage, we train it on 8 3090 GPUs (24G) for 5 days and for the fine stage, we train it on 8 A100 (80G) for 3 days.

### A.3 MORE RESULTS

### A.3.1 QUALITY RESULTS

**View consistency problem** We gives more view inconsistency visualization problem by visualize center of Gaussians from each view in different colors, as shown in fig. 7. Gaussians from different views representing the same part of the object may lays on the different position in the 3D space and thus cause the view inconsistency problem.

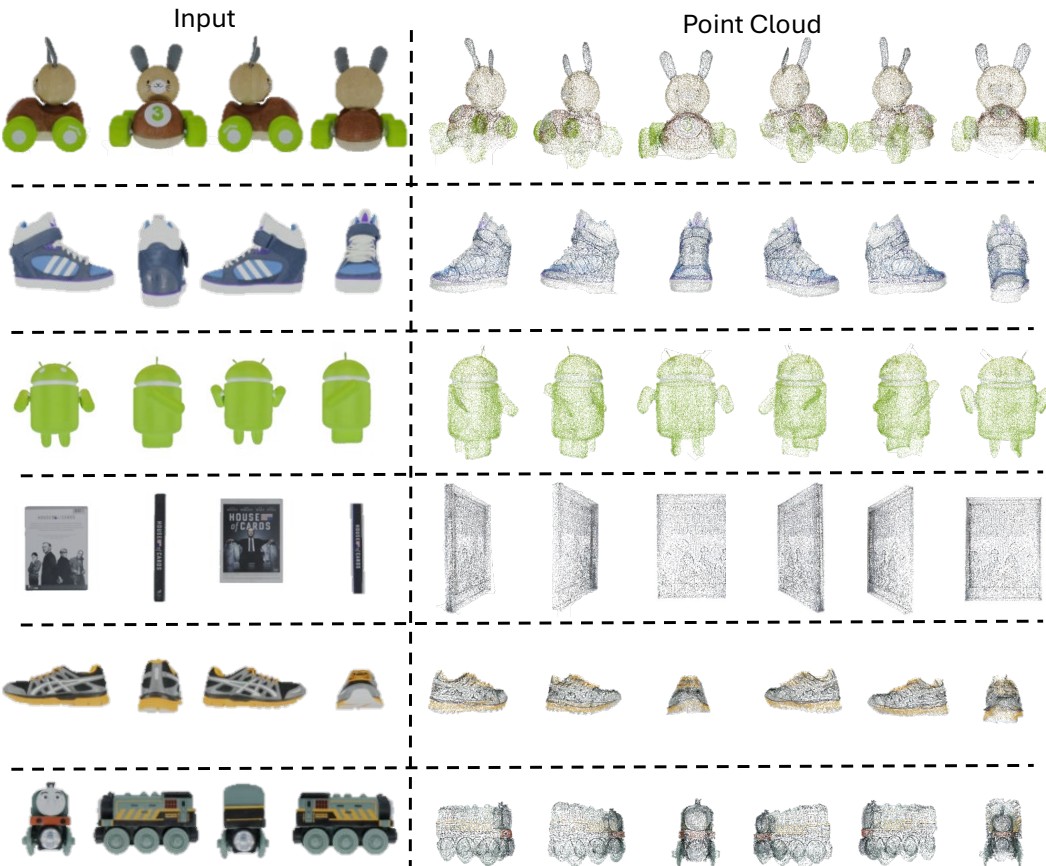

**Figure 8:** 3D Gaussian center as point cloud on GSO-fixed dataset for inputting 4 views.

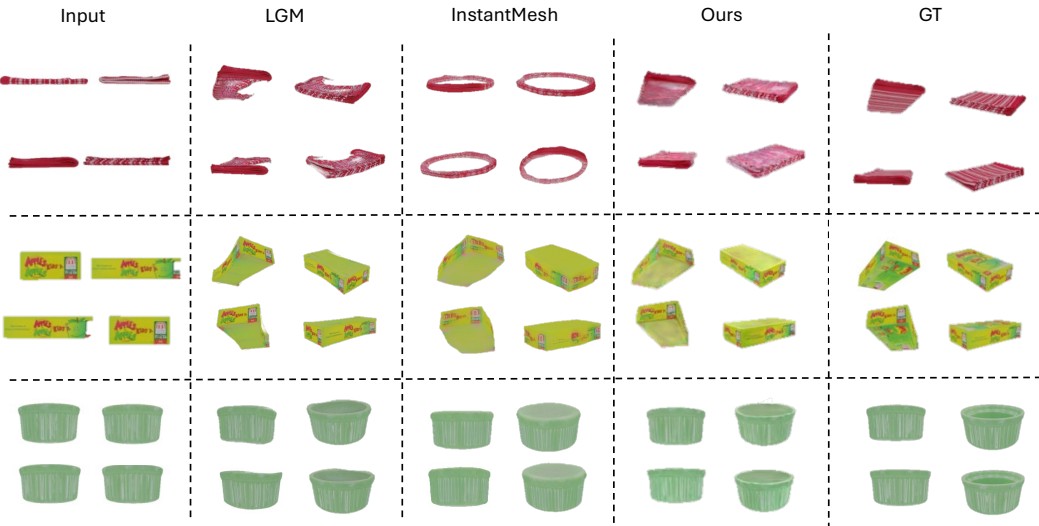

**Figure 9:** Quality for rendered novel views on GSO-fixed dataset for inputting 4 views with resolution 256 LGM large model.

**More visualization** We show the point cloud visualization in fig. 8 underscores our model's ability to capture geometry effectively, not just rendering quality.

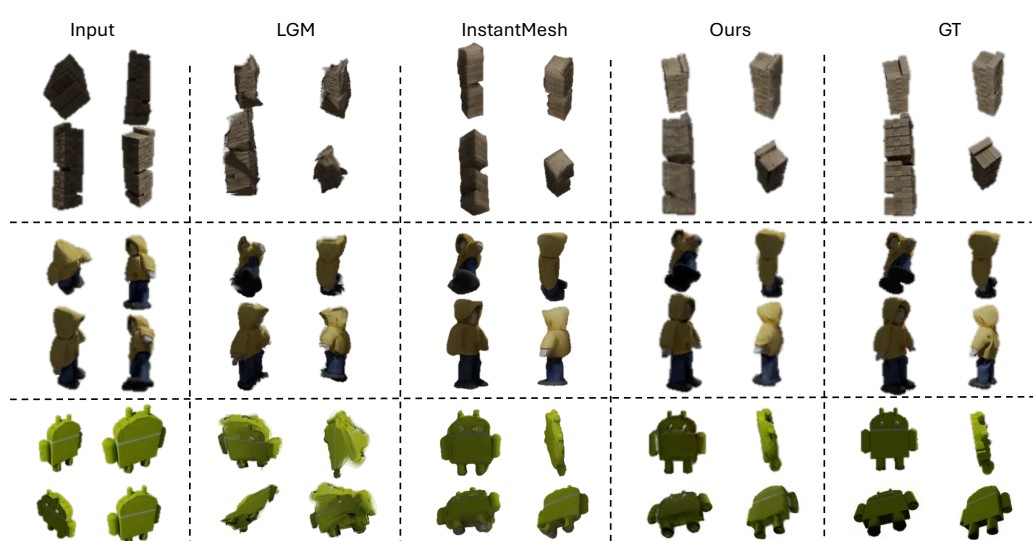

**Figure 10:** Quality for rendered novel views on GSO-random dataset for inputting 4 views.

As shown in fig. 9, when given limited number of input, neither LGM nor InstantMesh gives the meanful geomery.

fig. 10 presents the quantitative results of novel views rendered by recent models trained on 4 views. When provided with 4 random views as input, LGM (Tang et al., 2024a) demonstrates a loss of geometry and encounters view inconsistency problems stemming from its training on fixed views. In contrast, our approach produces a cohesive 3D Gaussian set that effectively captures object geometries.

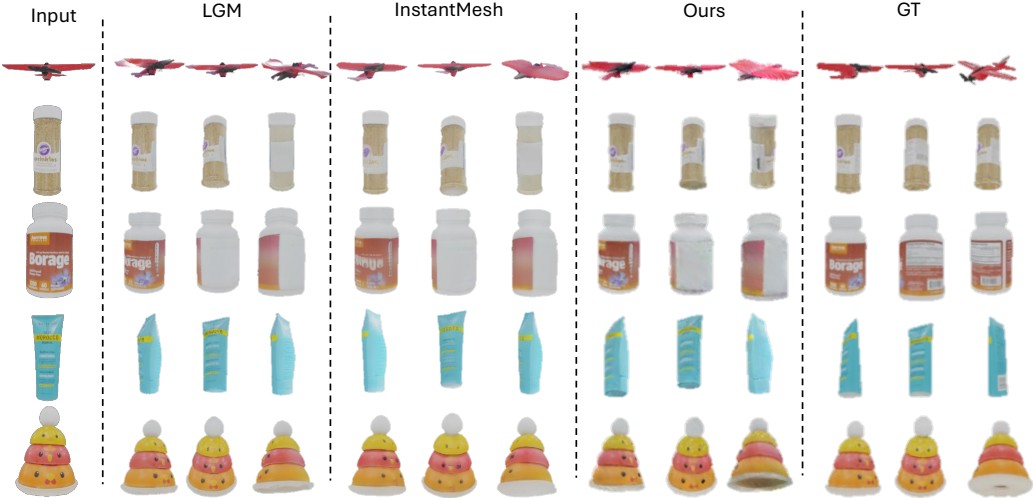

**Figure 11:** Quality for rendered novel views on GSO dataset for inputting 1 view and using Image-Fusion to generate 4 views.

fig. 11 and fig. 12, respectively. The figures illustrate that LGM encounters the issue of view inconsistency; for instance, there are multiple handles visible for the mushroom teapot. InstantMesh loses some details due to its utilization of a discrete triplane to represent continuous 3D space.

fig. 13 shows the result of text-to-3D task. We have incorporated text-to-3D capabilities into our model. To assess quality, we employ MVDream (Shi et al., 2024) to create a single image from a

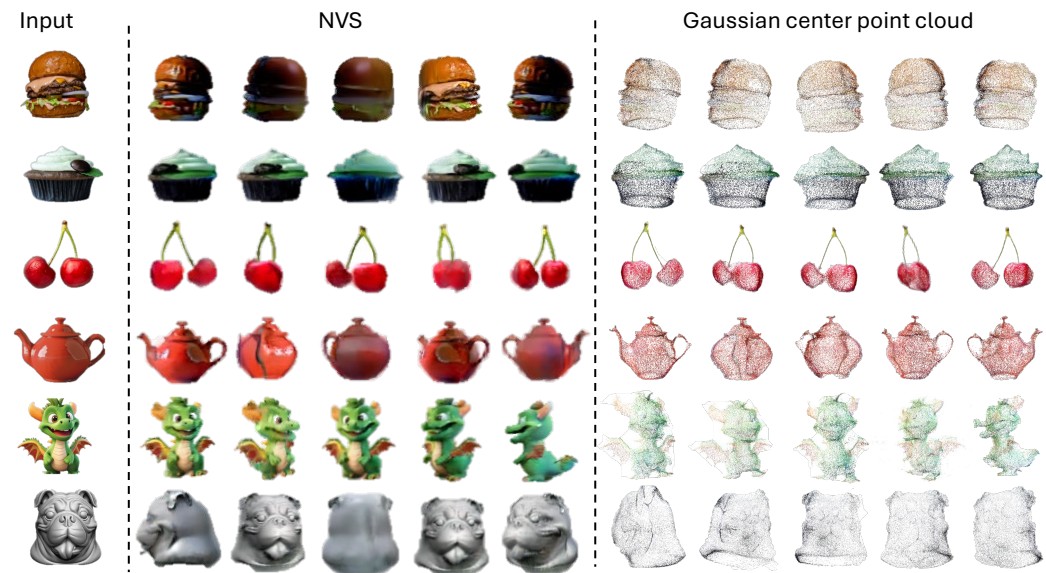

**Figure 12:** Quality for rendered novel views on in the wild data for inputting 1 view and using ImageDream to generate 4 views.

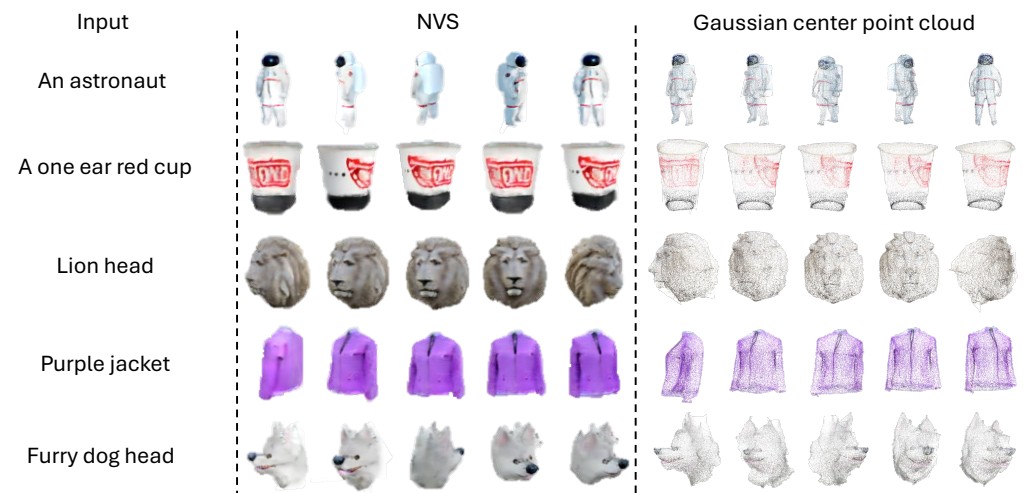

**Figure 13:** Quality for rendered novel views on inputting text and using MVDream to generate 4 views.

text prompt. Subsequently, a diffusion model is utilized to generate multi-view images, which are then processed by our model to obtain a 3D representation.

The setting of random input view is obvious a more challenging task than the setting of fixed input view, thus our method also inevitably suffers from a performance drop but still perform better than other state-of-the-art methods. As for Splatter Image (Szymanowicz et al., 2024), it also meets a significant performance drop when random input views are used as its SSIM ↑ decreased from 0.9151 to 0.8932 and LPIPS ↓ increased from 0.1517 to 0.2575 despite its PSNR ↑ has a slight increase. We visualize the results of the two settings to show the difference in fig. 14.

**Visualization with resolution 512** We provide the visualization result with resolution 512 in fig. 20.

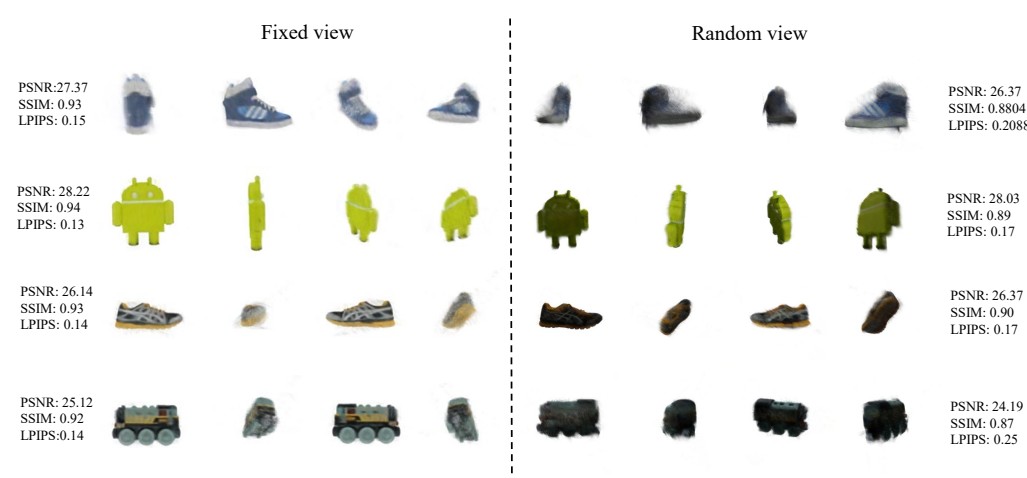

**Figure 14:** Visualization for Splatter Image with fixed view input and random view input.

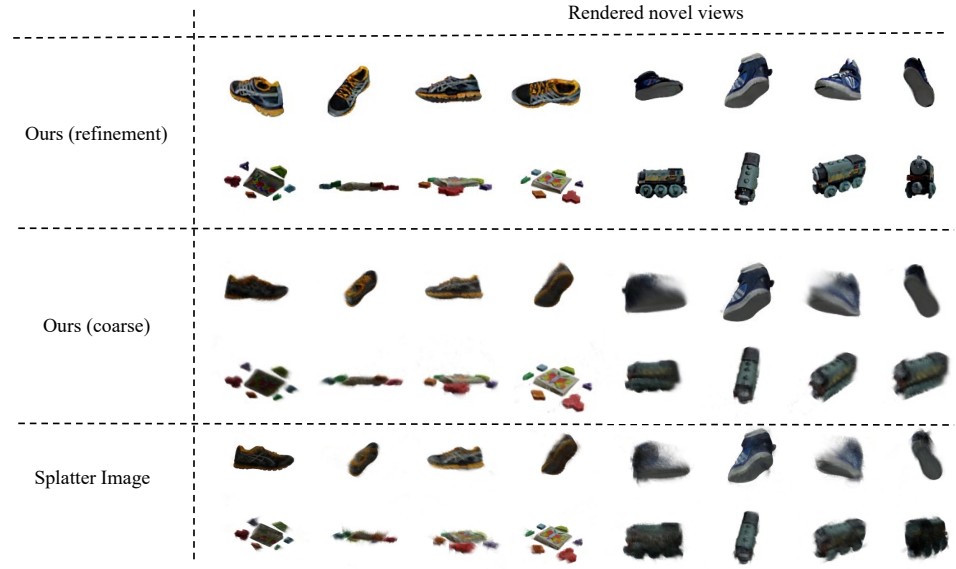

**Figure 15:** Visualization for Splatter Image with fixed view input and random view input.

### A.3.2 QUANTITY RESULTS

**Splatter Image visualization** PSNR of Splatter Image in table 2 is good but SSIM and LPIPS are not good enough, we further provide the visualization is in fig. 15.

Table 5: Quantitative results trained on Objaverse LVIS and tested on GSO. 3D sup. means need 3D supervision.

| Method | PSNR ↑ | SSIM ↑ | LPIPS ↓ | 3D sup. | Inference time | Rendering time |
|---|---|---|---|---|---|---|
| Triplane-Gaussian (Zou et al., 2024) | 18.61 | 0.853 | 0.159 | ✓ | 1.906 | **0.0025** |
| TripoSR (Tochilkin et al., 2024) | 20.00 | 0.872 | 0.149 | ✗ | 3.291 | 22.7312 |
| Ours | **23.45** | **0.897** | **0.093** | ✗ | **0.476** | **0.0025** |

**Single image reconstruction** There are common points between our model and TriplaneGaussian and Instant3D that we all use a unitary representation and use Transformer to regress. For

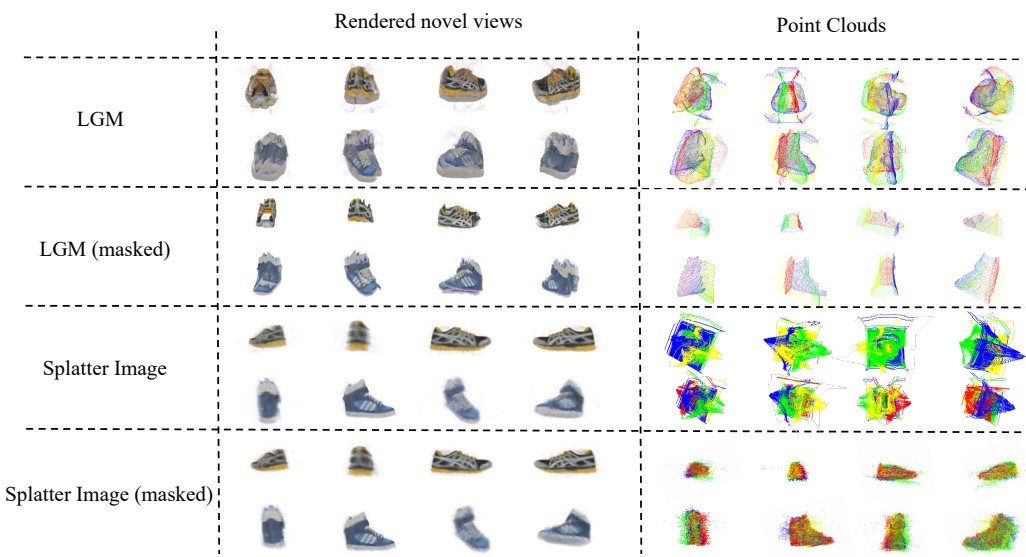

**Figure 16:** Removing the background use mask for Splatter Image and LGM

Instant3D, it transformers image to Nerf, making longer rendering time. For Triplane Gaussian, which is a single view reconstruction model with complex and costly triplane representation, representing compresses 3D space, leading to a lack of detailed information in the 3D structure and imposing a rigid grid alignment that limits flexibility (Tang et al., 2024a; Qi et al., 2017a). In the contrast, we use a more efficient way (deformable attention) to decode Gaussians. The comparison between Triplane-Gaussian and our methods is shown in table 5. Triplane Gaussian requires 3D supervision and takes longer inference time while get worse performance comparing to our model. We test on the given light-weight checkpoint in the github on the single view situation. We also test TripoSR (Tochilkin et al., 2024) on the single image reconstruction setting. As shown in table 5, our model surpass the previous methods on both the performance and the inference speed. We provide the visualization results of our model on single image reconstruction task in fig. 19

Table 6: Comparison between masked and original pixel aligned methods

| Method | PSNR ↑ | SSIM ↑ | LPIPS ↓ |
|---|---|---|---|
| LGM | 17.4810 | 0.7829 | 0.2180 |
| LGM (masked) | 21.6008 | 0.8608 | 0.1232 |
| Splatter Image | 25.6241 | 0.9151 | 0.1517 |
| Splatter Image (masked) | 25.0648 | 0.9147 | 0.1684 |

**Comparison to masked LGM and Splatter Image**    To better explain that the view inconsistency problem is not caused by the background points from previous methods, we provide the results on removing background points of LGM and Splatter Image. LGM uses mask loss to make the most of the pixels contribute to the object itself, even for the background pixels, therefore, removing background use mask makes the results more sparse. It also removing some outliers and thus the rendering results is better as shown in table 6. Splatter Image keep most of the pixels contribute to its original position, making most of the background points still located on a plane instead of the object. Therefore, removing background use mask does not influence the rendering result much but the rendering quality still reduced a little. Moreover, the view-inconsistency is not caused by the background points but the mis-alignment of 3D Gaussians from different views, removing the background use mask does not help solving the problem. We show the visualization in fig. 16

**Other number of view results**    We present the results of training with varying numbers of views (2, 6, 8) and evaluate the corresponding results with the same number of views in table 7.

Table 7: Quantitative results of novel view synthesis training using 2, 6, and 8 input views, tested on the GSO-random dataset across 2, 6, and 8 views.

| Method | 2 views | | | 6 views | | | 8 views | | |
|---|---|---|---|---|---|---|---|---|---|
| | PSNR ↑ | SSIM ↑ | LPIPS ↓ | PSNR ↑ | SSIM ↑ | LPIPS ↓ | PSNR ↑ | SSIM ↑ | LPIPS ↓ |
| Splatter Image | 22.6390 | 0.8889 | 0.1569 | 26.1225 | 0.9178 | 0.1620 | 26.4588 | 0.9166 | 0.1714 |
| Our Model | **23.8384** | **0.8995** | **0.1254** | **28.1035** | **0.9489** | **0.0559** | **28.8262** | **0.9537** | **0.0492** |

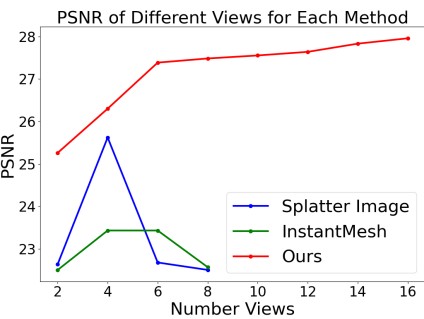

**Figure 17:** Visualization for Splatter Image with fixed view input and random view input.

Our model is positioned on the 'sparse view' setting, which indicates the number of views less then 10, so we only reports the performance of views from 2 to 8 in the main paper. With the increase of input views, information from similar views becomes redundant, so the gain for our model has become plateaued while other methods suffer from performance drop as they cannot handle too many input views due to the view inconsistent problem. As we keep increasing the number of input views larger than 8, our method can still benefit from more input views (as shown in fig. 17) while others meet the CUDA-out-of-memory problem.

**Image-to-3D**  Image-to-3D conversion represents a fundamental application in 3D generation. Following the methodology of LGM and InstantMesh (Tang et al., 2024a; Xu et al., 2024), we first leverage a multi-view diffusion model, ImageDream (Wang & Shi, 2023), to generate four predetermined views. Subsequently, our model is employed for 3D Gaussian reconstruction. A comparative analysis with LGM and InstantMesh is detailed in table 8. For this particular scenario, we utilize the fixed-view GSO test set with elevations ranging between 0 and 30 degrees. Given potential variations in camera poses among the generated multi-views, which may not align precisely with standard front, right, back, and left perspectives, we selectively retain 266 objects that consistently yield accurate images under the provided camera poses.

Table 8: Quantitative results for single view reconstruction on GSO dataset.

| Method | PSNR ↑ | SSIM ↑ | LPIPS ↓ |
|---|---|---|---|
| LGM (Tang et al., 2024a) | 20.8139 | 0.8581 | 0.1508 |
| InstantMesh (Xu et al., 2024) | 19.4667 | 0.8379 | 0.1842 |
| Our Model | **22.3534** | **0.8567** | **0.1492** |

### A.4  ABLATION STUDY

**Number of views in the coarse stage**  We add the ablation study on the number of images used during the coarse stage here. The results shown is that the number of images used during the coarse stage does not influence the final result. The reason that we choose the number of views being 2 is that we want to support any number of input views. For example, if we choose the number of views in the coarse stage being 8, we should at least provide 8 views so that the model can not support the number of views smaller than 8. And we tried to change the input views but the number of input views keeping 2 unchanged, the variance of PSNR for 10 different experiments is within 0.185.

**Convergence for different regression target**  Upon investigation, we observe that prior techniques frequently predict depth rather than the centers of Gaussians. In our exploration, we conduct experiments focusing on regressing the centers of 3D Gaussians while keeping other aspects constant. Through this analysis, we discover that regressing the positions of 3D Gaussians can in-

Table 9: Ablation study results of different view and different number of views for the coarse stage (with 4 views in the refinement stage)

| Number of views in coarse stage | PSNR ↑ | SSIM ↑ | LPIPS ↓ |
|---|---|---|---|
| 1 | 30.2312 | 0.9608 | 0.0413 |
| 2 | 30.4245 | 0.9614 | 0.0422 |
| 3 | 30.3442 | 0.9618 | 0.0419 |
| 4 | 30.4521 | 0.9620 | 0.0412 |

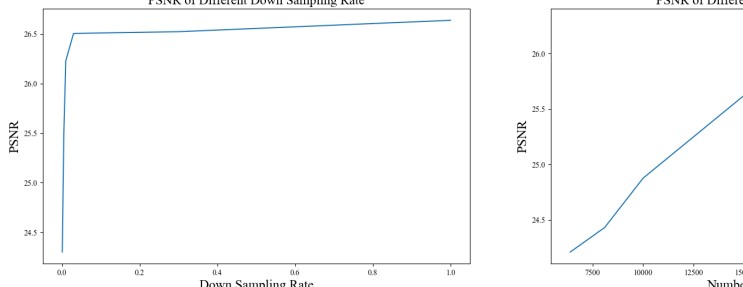

**Figure 18:** Left: PSNR with different down sampling rate in the spatial efficient self attention. Right: PSNR with different number of Gaussians.

troduce convergence obstacles. Table table 10 illustrates the outcomes of these experiments on the Objaverse validation dataset after 100K steps.

Table 10: Ablation study on parameter selection.

| Regression target | PSNR ↑ | SSIM ↑ | LPIPS ↓ |
|---|---|---|---|
| Depth | 24.3792 | 0.9012 | 0.1014 |
| 3D Gaussian centers (random initialize in visual cone) | 19.2551 | 0.8343 | 0.1876 |
| Coarse-to-fine | **25.5338** | **0.9126** | **0.0833** |

**More ablation studies**  Here we gives more ablation study mainly for hyperparameter selection. Due to computational costs, ablation models are trained at 100k iteration and test on Objaverse validation dataset.

**Hyperparameter selection**  As previously highlighted, the memory bottleneck of our model lies in the pointwise self-attention mechanism. To address this, we implement a spatially efficient self-attention technique to alleviate memory consumption. Illustrated in fig. 18 (left), as we augment the downsampling rate of the key and value in the self-attention mechanism, the memory overhead diminishes linearly, while the PSNR reduction is not as rapid. Consequently, we opt for a down-sampling rate located at the inflection point, which we determine to be 0.01, balancing memory efficiency with reconstruction quality. Similarly, we select the number of Gaussians as 19600 as shown in fig. 18 (right).

In table 11, we opted for 4 decoder layers over 6, as the latter offers marginal improvement but demands significantly more computational resources. Additionally, we experimented with using the fine stage initialized with the coarse stage as the encoder and tested the efficacy of fine-tuning both stages. Our findings indicate that fine-tuning both stages yields the best results.

A.5   COMPARISON TO MVSPLAT AND PIXELSPLAT

We present a comparative analysis involving MVSplat (Chen, Yuedong and Xu, Haofei and Zheng, Chuanxia and Zhuang, Bohan and Pollefeys, Marc and Geiger, Andreas and Cham, Tat-Jen and

Table 11: Ablation study on parameter selection.

| Method | PSNR ↑ | SSIM ↑ | LPIPS ↓ |
|---|---|---|---|
| 2 decoder layers | 24.5229 | 0.9195 | 0.1021 |
| 6 decoder layers | **26.2442** | **0.9352** | **0.0778** |
| Freeze coarse stage finetune encoder | 25.6902 | 0.9223 | 0.0826 |
| Freeze both coarse stage and encoder | 25.3211 | 0.9264 | 0.1003 |
| Default model | 26.2313 | 0.9351 | 0.0788 |

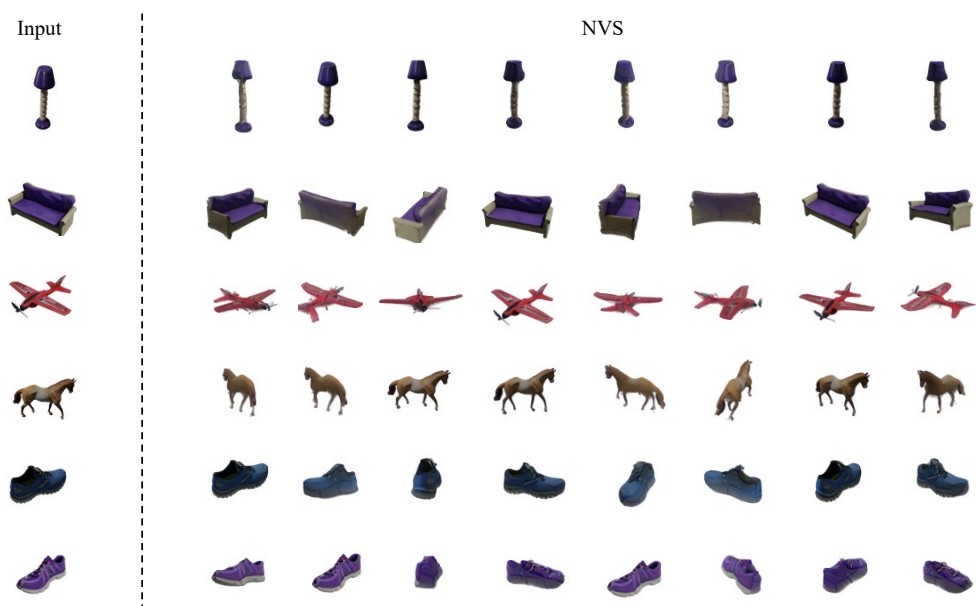

**Figure 19:** Single view 360 rendering visualization on GSO dataset

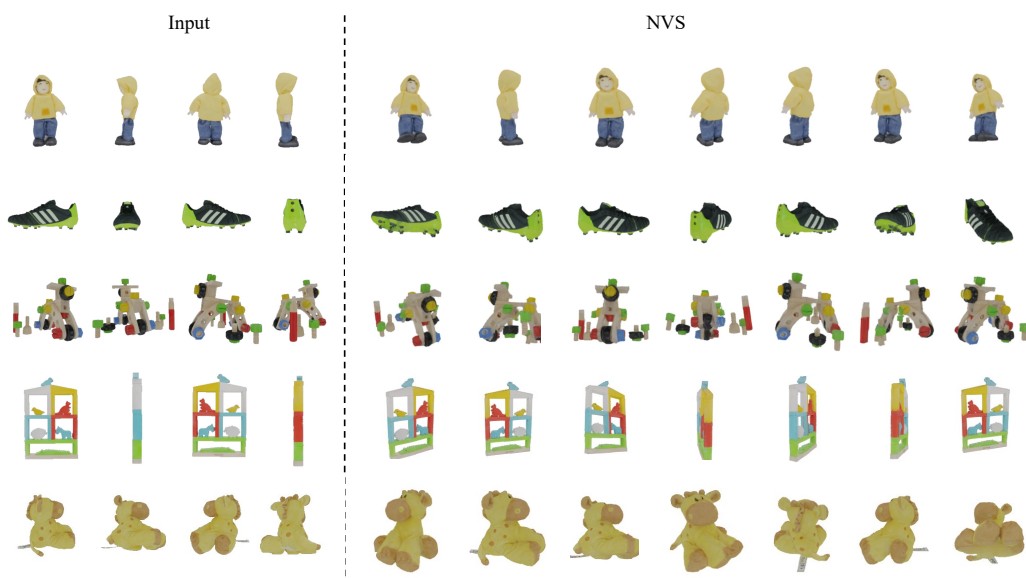

**Figure 20:** Visualization for our method with resolution 512

Cai, Jianfei, 2024) and pixelSplat (Charatan et al., 2024) on the GSO-random dataset using the provided checkpoints from the repository in this section. Similar to LGM (Tang et al., 2024a), both aforementioned methods follow a workflow that regress Gaussians from each views within the respective camera spaces and subsequently merge them in the global world space. Despite pixelSplat's integration of cross-view-aware features through an epipolar Transformer, accurately forecasting a dependable probabilistic depth distribution based solely on image features remains a formidable task (Chen, Yuedong and Xu, Haofei and Zheng, Chuanxia and Zhuang, Bohan and Pollefeys, Marc and Geiger, Andreas and Cham, Tat-Jen and Cai, Jianfei, 2024). This limitation often translates to pixelSplat's geometry reconstruction exhibiting comparatively lower quality and plagued by noticeable noisy artifacts (Chen, Yuedong and Xu, Haofei and Zheng, Chuanxia and Zhuang, Bohan and Pollefeys, Marc and Geiger, Andreas and Cham, Tat-Jen and Cai, Jianfei, 2024). Upon examination, we observed that even after isolating points within a visual cone and eliminating background Gaussians, the geometry fails to convey meaningful information, yielding unsatisfactory results.

In contrast, MVSplat adopts a design that incorporates a cost volume storing cross-view feature similarities for all possible depth candidates. These similarities offer crucial geometric cues for 3D surface localization, leading to more substantial depth predictions. However, akin to Splatter Image, which assigns each pixel a Gaussian and thereby generates a planar representation rather than the object itself, MVSplat's approach may obscure object details due to occlusion by background Gaussians from other viewpoints, resulting in suboptimal outcomes.

To address this issue, we selectively mask the positioning of Gaussians on background pixels, focusing solely on rendering Gaussians contributing to the object itself. This adjustment reveals significant view inconsistency problems, as illustrated in fig. 21. In the figure, we present the centers of Gaussians generated from different views in different color and the novel views are rendered from the Gaussians from all views. Furthermore, the elaborate incorporation of cross-view attention mechanisms and cost volumes in MVSplat leads to extended inference times and heightened memory requirements as shown in table 12.

Table 12: Comparison with MVSplat and pixelSplat on the GSO-random dataset in the 4-view input setting.

| Method | PSNR ↑ | SSIM ↑ | LPIPS ↓ | Inference time | Rendering time |
|---|---|---|---|---|---|
| MVSplat | 12.92 | 0.80 | 0.30 | 0.112 | 0.0090 |
| MVSplat (masked) | 16.52 | 0.80 | 0.19 | 0.112 | 0.0045 |
| pixelSplat (2 views) | 12.00 | 0.80 | 0.28 | 1.088 | 0.0045 |
| pixelSplat (2 views masked) | 12.05 | 0.79 | 0.27 | 1.088 | 0.0023 |
| Ours | **26.30** | **0.93** | **0.08** | **0.694** | **0.0019** |

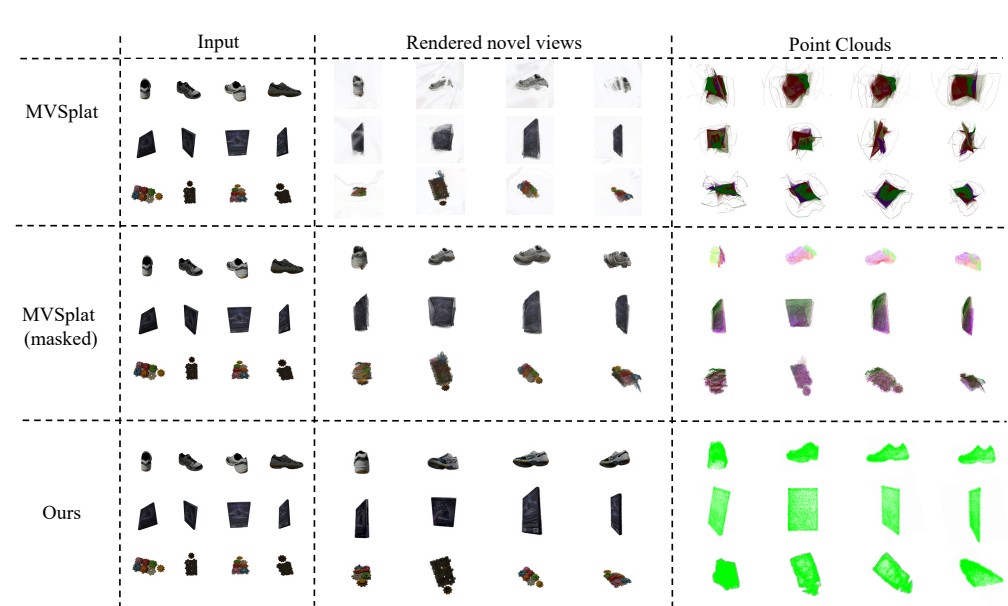

**Figure 21:** Visualization for MVSplat and our method

