# OpenReview forum: "UniG: Modelling Unitary 3D Gaussians for View-consistent 3D Reconstruction"
_ICLR.cc/2025/Conference — Submitted to ICLR 2025_

### Official Review · Reviewer_t736 · 2024-11-02

**Soundness:** 3
**Presentation:** 3
**Contribution:** 3
**Rating:** 5
**Confidence:** 5

**Summary:**

The paper introduces a feed-forward method for 3D object reconstruction from sparse input views, utilizing 3D Gaussian Splatting (3D GS) as its scene representation. This method predicts a unified set of 3D Gaussians from multiple input images rather than generating per-view, pixel-aligned Gaussians as in previous approaches. This approach utilizes a DETR-like transformer framework, treating 3D Gaussians as decoder queries and updating their parameters through multi-view cross-attention layers. The results on several benchmark datasets demonstrate promising quality and better than some previous pixel-aligned approaches.

**Strengths:**

A key strength of this paper lies in its application of a DETR-like transformer framework to predict an independent unitary set of 3D Gaussians for 3D reconstruction. This method shows promise in applying such a network architecture to solve the sparse-view reconstruction problem, leading to promising results.

**Weaknesses:**

1. There are multiple highly relevant prior works that are neither cited nor discussed in the paper. This includes works on per-pixel Gaussian splatting prediction, such as
    * GS-LRM: Large Reconstruction Model for 3D Gaussian Splatting
    * GRM: Large Gaussian Reconstruction Model for Efficient 3D Reconstruction and Generation
    * pixelSplat: 3D Gaussian Splats from Image Pairs for Scalable Generalizable 3D Reconstruction
    * MVSplat: Efficient 3D Gaussian Splatting from Sparse Multi-View Images

    Additionally, other works on sparse-view 3D reconstruction and generation are also absent, including
    * Instant3D: Fast Text-to-3D with Sparse-View Generation and Large Reconstruction Model
    * One-2-3-45++: Fast Single Image to 3D Objects with Consistent Multi-View Generation and 3D Diffusion
    * MeshLRM: Large Reconstruction Model for High-Quality Mesh
    * MeshFormer: High-Quality Mesh Generation with 3D-Guided Reconstruction Model
    * TripoSR: Fast 3D Object Reconstruction from a Single Image.

    Without proper citation and discussion of these papers, I am highly concerned about the positioning of the proposed approach.

2. The paper criticizes the design of per-pixel Gaussian prediction in previous works (LGM, MVGamba, etc), attributing it to the low quality and view inconsistency issues. However, this argument overlooks the success of other per-pixel methods listed above, such as GS-LRM and GRM, which demonstrate high-quality, view-consistent results, which, to me, look even more visually realistic than the results shown in the paper. The paper compares primarily against weaker baselines like LGM (or others with similar or lower quality), showing improvements over LGM. However, I've seen many existing papers that have demonstrated significantly better quality than LGM, including GS-LRM, GRM, Mesh-LRM, MeshFormer, as listed above. In general, the paper fails to compare with or even discuss these stronger, state-of-the-art methods. In particular,  GS-LRM and GRM also employ per-pixel strategies yet seem to achieve even greater improvements when seeing their results in their paper and website. This kind of suggests that the design choice of per-pixel prediction may not be the main issue with the baselines (like LGM) discussed in the paper, and that there could be the other architectural factors in those models that led to the lower quality. As a result, the paper's argument for its method being inherently superior to per-pixel techniques is less convincing.

**Questions:**

While I think the unitary Gaussian prediction technique in the paper is promising, I am really concerned about the paper's positioning due to the absence of numerous relevant prior works. I understand that direct comparisons can be challenging, especially given that many of these previous works have not released code. However, at minimum, a thorough discussion is needed to place this work in context, and any feasible comparisons would significantly strengthen the paper. I feel, at least, the paper needs to tone down and moderate its claim by incorporating and discussing all these relevant works properly.

---

> ### Author Response · Authors · 2024-11-20
>
> Thank you very much for your thoughtful comments on our work. We greatly appreciate your feedback. We summaries the main concerns in weaknesses1 and 2 and response them in the following.
> We agree that these related methods should be mentioned in our paper to position our paper in the field of 3D reconstruction. We discuss each method below and have added them in the related work part in the revised PDF.
> We will add the quantitative comparison with those that open-sourced codes are available as soon as possible. Should you have any further questions or concerns, please do not hesitate to reach out to us.
>
> # Weakness
> We summaries the main concerns in weaknesses1 and 2 and response them in the following. For GS-LRM and GRM, by merely reading their papers, both of them claim that they use a scalable large transformer with the similar pixel-aligned structure and are trained with many A100 GPUs but it is not clear what are the key components that has made their model work well.
> Moreover, the view inconsistent problem theoretically exists in all those methods that first predict 3D Gaussians in each camera space and then naively merge them in world space as the depth of the predicted 3D Gaussians in each view would always have errors, which will inevitably lead to the misaligned 3D Gaussians merged in the world space. The problem is clearly illustrated in the attached video inconsistentpc.mp4.
>
> However, we are not able to fairly compare our method with them as they have not released the codes. And their quantitative results are computed not following the same dataset or settings with previous methods, making them hard to be fairly compared quantitatively. For MVSplat and pixelSplat, we are now working on comparing them and will release the comparison results as soon as possible. For One-2-3-45++, it is a generative model, suffering longer inference time that take 20 seconds to 1 minute for one generation while fast-forward methods only takes around 1 second. As for Mesh-LRM and MeshFormer, they use representations like NeRF and voxel, which have disadvantages in the aspect of computational efficiency, especially for rendering. Moreover, MeshFormer requires 3D supervision while Mesh-LRM utilize triplane, who compresses 3D space, leading to a lack of detailed information in the 3D structure and imposing a rigid grid alignment that limits flexibility ([1][2]). TripoSR is a single image 3D reconstruction model, which uses an encoder-decoder structure with triplane decoder and triplane-based NeRF, and it also has the limitations of NeRF and Triplane as discussed earlier. Its inference time is 3.29s for a single forward process (without mesh extracting) and rendering time for 22.73s, which is much slower than 3D GS-based models. We added the discussion of the mentioned methods in the related work part of the revised version and will add the comparison with MVSplat and pixelSplat as soon as possible.
>
> # Questions1:
> The main contribution of our method is proposing a new unitary 3D Gaussian modeling approach for multi-view 3D reconstruction that can avoid the view inconsistent problem. This new design can also be adopted by other mentioned 3D GS-based methods. We will discuss more about the mentioned prior works in the revised PDF.
>
> [1] Jiaxiang Tang, et al., LGM: Large Multi-View Gaussian Model for High-Resolution 3D Content Creation. ECCV2024.
>
> [2] Charles R Qi, et al., PointNet: Deep Learning on Point Sets for 3D Classification and Segmentation. In CVPR 2017a

---

> ### Author Response · Authors · 2024-11-22
> **Add results of TrioSR on GSO dataset for single view situation**
>
> # Results of TrioSR on GSO dataset for single view situation
>
> We test TripoSR and Triplane-Gaussian on the single image reconstruction setting with the checkpoint they provide on github. As shown in the following table, our model surpass the previous methods on both the performance and the inference speed. In terms of rendering speed, both Triplane-Gaussian and our model employ Gaussian Splatting, known for its fast rendering speed. Conversely, TripoSR utilizes NeRF, a method slower in comparison to Gaussian Splatting.
>
> **Table 1:** Quantitative results trained on Objaverse LVIS and tested on GSO. 3D sup. means need 3D supervision.
> | Method                               | PSNR $\uparrow$ | SSIM $\uparrow$ | LPIPS $\downarrow$ | 3D sup.    | Inference time | Rendering time |
> |--------------------------------------|-----------------|-----------------|--------------------|------------|----------------|----------------|
> | Triplane-Gaussian | 18.61           | 0.853           | 0.159              |✔ | 1.906          |  **0.0025**  |
> | TripoSR            | 20.00           | 0.872           | 0.149              |✘ | 3.291          | 22.7312        |
> | Ours                                 | **23.45**       | **0.897**       | **0.093**          | ✘          | **0.476**     | **0.0025**    |
>
> We have updated this table as Table 5 in the revised version of the PDF.

---

> ### Comment · Reviewer_t736 · 2024-11-24
>
> I appreciate the authors' response and the additional experiments provided.
>
> However, I'm not convinced that "the view inconsistency problem theoretically exists" in all per-pixel Gaussian prediction methods. While it is true that "the depth of the predicted 3D Gaussians in each view would always have errors," the authors’ method will also introduce errors, albeit not view-aligned. In general, those per-view Gaussian methods are designed to learn to aggregate multi-view Gaussians in an end-to-end manner, which learns to fuse the depth and achieve consistency. Theoretically, as these methods minimize errors during training, their consistency and quality improve. In essence, both per-view Gaussian methods and the proposed unitary method rely on similar processes of network optimization and gradient descent to achieve consistency and rendering quality. Of course, the proposed method doesn't have a concept of "view-"consistent depth, but it has 3D errors of point locations, which might also lead to inconsistent renderings across novel viewpoints. So I cannot agree that this is a theoretical limitation.
>
> Regarding the inconsistencies shown in inconsistentpc.mp4 and other supplementary examples, I revisited the GRM and GS-LRM websites to verify their results. GS-LRM, for instance, provides a lot of ply files of their GS reconstruction shown in an interactive 3DGS viewer. I reviewed several examples in their viewer and also downloaded multiple ply files to inspect in Meshlab, finding their results to be of high quality and very consistent. I did not observe a similar level of inconsistencies shown in the LGS examples in the supp.
>
> Overall, I do not believe the inconsistency issue is a theoretical problem inherent to all per-view Gaussian methods. The evidence suggests that this issue may be unique to LGM or other baselines. To clarify, I do not hold a strong bias toward per-view Gaussian methods, and I am also disappointed about papers like GS-LRM not releasing their code to facilitate comparisons. I also really appreciate the authors' efforts to explore unitary GS prediction, but my main concern is the potential to mislead the community here. The paper currently is written like per-view Gaussian methods are inherently worse, which I do not see sufficient evidence to support. If the inconsistency problem is specific to LGM or other baselines, this should be clearly stated, and the issue should not be generalized to all per-view Gaussian methods.
>
>
> On the other hand, the new single-image reconstruction results are very impressive. However, I am surprised by the reported PSNR values, as they seem unusually high. To be more specific, I checked the numbers in the recent NeurIPS paper MeshFormer; the SOTA single-view reconstruction methods achieve PSNRs only around 21 but their quality already looks very good. Typically, the PSNR for single-image reconstruction is low because the unseen back side of an object introduces significant uncertainty, making it theoretically impossible to recover accurately. Deterministic models often produce blurry back-side renderings, while probabilistic models, such as diffusion models, tend to generate sharp but potentially mismatched back-side renderings. I personally even feel PSNR and other rendering metrics are not the best choice to evaluate single image reconstruction techniques because this is more of a generative task. But why can the proposed method lead to such a high PSNR over 23? Could the authors clarify how this experiment was conducted? Specifically, details on the resolution, training/testing view settings, and other relevant parameters would be helpful. Additionally, providing more visual examples, including input views and 360-degree renderings, would help illustrate what is happening in the reconstructions.

---

> > ### Author Response · Authors · 2024-11-25
> >
> > Thanks for your feedback and we really appreciate it.
> >
> > We also checked the results of GRM and GS-LRM provided in their websites, and did not see obvious view-inconsistent reconstruction either. Thus, we agree that using the word ``all" may not be appropriate and we will avoid using it in our main paper. However, only from their presented good examples to totally negate our analysis is unfair and not reasonable as it also cannot be concluded that they do not have the problem. We are not able to analyze more cases (especially bad cases) of them because they have not released their codes, but we tested other open-sourced per-view Gaussian methods, including pixelSplat [3] and MVSplat [4], where we also observe the obvious view inconsistency problem as shown in the supplementary video mv\_splat.mp4. (Due to the supplementary size limitation, we only present one example while we actually observed other examples with the same problem.) Furthermore, we are sure that the simple 3D Gaussian merging of per-view methods are very likely to lead to the misaligned 3D Gaussians (i.e. view inconsistency) because the prediction of z-axis (depth) of 3D Gaussians in each view is an ill-posed problem whose error cannot be totally avoided. The problem maybe alleviated to some extent with proper gradient-based training but can never be avoided as long as the simple 3D Gaussian merging is still there. In some cases, the issue may not that severely be observed, but it may actually exist. In contrast, although our method is also not perfect, our proposed unitary modeling method principally avoid the ill-posed single-view z-axis (depth) prediction and the step of simple 3D Gaussian merging. Moreover, our method has cross-view information aggregation to comprehensively determine the update of unitary 3D Gaussians in the refinement stage. Both qualitative and quantitative results can validate the superiority of our method.
> >
> > In the single-image setting, our method already achieves a notable PSNR of 21.74 at the coarse stage. Following multi-layer refinement in the subsequent stage, the PSNR surpasses 23. We add visualization results in the revised PDF, showcased in Figure 19, illustrating input views and 360-degree renderings.
> >
> > The train and test details are as follows. For both training and testing phases, we conduct experiments at a resolution of 128. Consistent with our initial paper, we train on the Objaverse LVIS dataset and test on the GSO dataset. Notably, our model is trained on fixed number of views and can input any number of views during inference.   Testing aligns with the strategy outlined in Figure 5, where the model trains on 4 views and is subsequently tested with a single view, evaluating against the remaining 24 views. We use a 4-layer decoder in the refinement stage with each layer regressing the refinement of Gaussian parameters.
> >
> > Single-image approaches like InstantMesh [1] and MeshFormer [2] require existing diffusion models to generate multi-view images from a single image, then derive the final reconstruction results in a multi-view reconstruction manner. This pipeline can yields visually appealing visualizations, but the PSNR values tend to be lower due to distortions introduced by generative models. Our presented results have not follow this setting. The results following this setting are shown in Figure 11 and Table 8 of the main paper, where the PSNR of our method is 22.35.
> >
> >
> >
> > [1] XU, et al., InstantMesh: Efficient 3D Mesh Generation from a Single Image with Sparse-view Large Reconstruction Models. arXiv2404
> > [2] Liu, et al., MeshFormer: High-Quality Mesh Generation with 3D-Guided Reconstruction Model. NeurIPS 2024
> > [3] Charatan, et al., pixelSplat: 3D Gaussian Splats from Image Pairs for Scalable Generalizable 3D Reconstruction. CVPR2024
> > [4] Chen, MVSplat: Efficient 3D Gaussian Splatting from Sparse Multi-View Images. ECCV2025

---

> > > ### Comment · Reviewer_t736 · 2024-11-26
> > >
> > > I appreciate the authors' efforts to address the concerns raised and improve the paper's positioning.
> > >
> > > After reviewing the current draft, many sections now look fine. However, I still find the description of GS-LRM and GRM in the related work section problematic, particularly regarding the claim of a "theoretical problem." At a conference like ICLR, with a significant audience from the machine learning community, a theoretical claim is a very serious statement and typically requires a formal proof. Specifically, a proof that per-view methods theoretically lead to a higher lower bound of geometry error compared to the proposed method is needed if a theoretical claim must be made, which I don't feel you can really provide here. Currently, this discussion remains at an empirical level, not a theoretical one.
> > >
> > > Relevant to this, I’m not sure if the authors fully understand how GS-LRM and similar methods work. The paper and rebuttal repeatedly state that those methods predict depth independently, whereas the proposed method uses cross-view information. However, GS-LRM, for instance, employs full attention across all views during prediction, inherently incorporating cross-view information as well, which cannot be trivially seen as an independent prediction. This design also could explain why GS-LRM does not exhibit the view inconsistency. In fact, several other cited methods in the paper also adopt similar designs, so I don't feel the claimed independent prediction and cross-view information are really unique to the paper.
> > >
> > > Overall, I have the remaining concerns:
> > >
> > > 1. Paper positioning about pre-view vs unitary GS prediction as mentioned above.
> > >
> > > 2. Result Quality: While I understand that absolute fair comparisons are difficult due to the lack of code release, I do not find the results in the paper to surpass prior state-of-the-art methods like GS-LRM. I have compared many examples from the submission with results on the GS-LRM/GRM websites. Since all methods provide GSO results, it is easy to find the same or similar objects. I can easily see results from GS-LRM have sharper details than yours. I note your previous rely says "However, only from their presented good examples to totally negate our analysis is unfair and not reasonable as it also cannot be concluded that they do not have the problem." But in fact, I found more than 30 GSO results with the interactive viewer and downloadable plys from GS-LRM, which is even more than the total number of results shown in your paper and also overlaps with one or two of your examples. So I don't see potential concerns about cherry-picking here. On the other hand, the paper has few video/3d demonstrations, almost only showing a single result video that combines multiple results at a low resolution. If the authors can provide separate videos or viewable plys, it will be much easier for me and others to view and justify the quality.
> > >
> > > However, while I still have concerns about the quality, I can see the paper does show better results, especially under the fixed-view setting, than other baselines like LGM and MVGamba that are also recently published works. Therefore I won't see the quality issue as a strong blocker here. Since the paper also contributes a new reconstruction pipeline that is different enough from all previous ones, I overall feel the paper is on the bar of acceptance. So I would not object if other reviewers advocate for its acceptance. However, I am not ready to raise my score, as the writing/positioning issues have not been fully addressed. And I do not expect the authors to resolve my concerns about the quality since GS-LRM’s results are obviously superior qualitatively...

---

> > > > ### Author Response · Authors · 2024-11-27
> > > >
> > > > Thanks for your feedback and for recognizing our efforts.
> > > >
> > > > After reading your comments, we agree that we are not able to strictly prove the existence of the issue although we may logically infer it probably exists, so we modified our paper to avoid the expression of 'theoretical problem'.
> > > >
> > > > We know that other methods also incorporate multi-view information but they do this mainly in the early feature extracting stage by simply concatenate all image tokens to do self-attention, like GS-LRM. In this way, the geometry relationships among input views are mainly learned via a black-box manner, which may increase the learning difficulty (trained 2 days on 64 A100, while in our situation, trained for 3 days on 8 A100).
> > > > And after that, they predict 3D Gaussians in each view's camera space separately without other cross-view interaction and finally naively merge the multiple set of 3D Gaussians in the world space. Thus, we state 'predict depth independently' was to underscore that they would separately predict multiple set of 3D Gaussians in each input views' camera spaces rather than a unitary set. We have modified the corresponding statements in our main paper to avoid misunderstanding. In this point, the main difference of our method comparing to previous methods is that, in our method, all views collectively contribute to a unitary set of 3D Gaussians, instead of predicting multiple sets separately and then merging them. By projecting the unitary 3D Gaussian centers onto each view, the cross-view interaction module of our method incorporates the explicit geometry relationship between 3D Gaussians and 2D images to facilitate the learning of multi-view feature fusion.  We have made the adjustment in our related work section and present the revised version in the following.
> > > >
> > > > # Revised version in the related work:
> > > > "Various techniques such as SplatterImage, LGM, pixelSplat, and MVSplat have extended the application of 3D Gaussian Splatting to multi-view scenarios.
> > > > In these approaches, each input view is processed to estimate 3D Gaussians specific to the view, followed by a simple concatenation of the resulting 3D Gaussian assets from all views. GS-LRM and GRM exhibit a model structure similar to LGM, resulting in notable accomplishments through enhanced training processes and consequently more precise depth regression. Nevertheless, these models adhere to the pipeline of predicting 3D Gaussians separately for each view, they demands substantial computational resources, particularly as the number of views grows, the number of Gaussians scales linearly with the number of views. Furthermore, these methods are unable to accommodate an arbitrary number of views as input."
> > > >
> > > > # More results:
> > > > As for the concern of the quality, we presented more separate videos for you to check (Supplementary separate\_videos folder). We also provide a new visualization with resolution 512 in the revised PDF Appendix Figure 20. Compared with the results provided by GS-LRM in their website, our results are not that good but comparable, and are better than other baseline methods. We will try to figure out the behind reason of GS-LRM's good results and add the corresponding analysis as they release their codes. Given the description in the paper of GS-LRM, supposing they use the image resolution of $512\times512$, then there will be $512\times512\times4=1048576$ 3D Gaussians to reconstruct a single object in the 4-view setting while our method only use a fix number of $19600$ 3D Gaussians in consideration of the balance of computational resources for training and inference.

---

> > > > > ### Comment · Reviewer_t736 · 2024-12-02
> > > > >
> > > > > I appreciate the authors' continued efforts to address concerns regarding the writing and positioning of the paper.
> > > > >
> > > > > While I still have some disagreements with the current positioning, I think that my major concerns in this regard have been mostly addressed. It is also helpful that the authors clarified the use of substantially fewer Gaussians compared to prior work. While this design choice benefits efficiency, it likely contributes to the visual quality being somewhat inferior to GS-LRM. In my view, if the current method cannot scale to support significantly more Gaussians, this could represent a limitation, particularly for its potential extension to scene-level reconstructions requiring a larger number of Gaussians. That said, I consider this a minor point.
> > > > >
> > > > > On the other hand, I find the newly posted quantitative results disappointing and problematic. When comparing your method with Pixelsplat or MVSplat, I would expect you to either train your model on their training dataset or re-train their models on your data to ensure a fair comparison, but it looks like this is not what is done here. Objaverse and RealEstate10K are fundamentally different datasets—one focusing on object assets and the other on real-world scenes—so it is unsurprising that models trained on one dataset fail to generalize well to the other. As a result, the new experiment does not provide meaningful insights. I suggest either removing the experiment or redoing it properly (though the latter may not be feasible within the current timeline).
> > > > >
> > > > > I usually hate raising new concerns at the last moment, and I should have checked your response earlier. But I really feel including these results in their current state would be misleading and reduce the overall quality of the paper.

---

> > > > > > ### Author Response · Authors · 2024-12-02
> > > > > >
> > > > > > Thank you for your feedback and acknowledgment of our efforts. We are glad to hear that your major concerns have been mostly addressed.
> > > > > >
> > > > > > For the number of 3D Gaussians to use in our method, we have plotted a figure in Figure 18 of the main paper. In the figure, we presented the ablation study results on the impact of the number of 3D Gaussians. With the increase of 3D Gaussians' number, the PSNR on the test set also increase almost linearly until it reach
> > > > > > the number of around 19600. Thus, we chose this number to conduct the other experiments to achieve a balance between performance and computational efficiency. If we keep raising the 3D Gaussians' number, our method's PSNR on the test set continues to grow, albeit not as rapidly. Thus, we can generally conclude that our method can also benefit from the scale-up of the number of 3D Gaussians.
> > > > > >
> > > > > > For the experiments on the pixelSplat and MVSplat, we regret what we have done is not as you expect. We will remove this part or add the results of them after training on the Objaverse dataset, following your suggestion. We now have successfully run the training code of them on the Objaverse dataset and everything looks correct. We will keep checking the correctness of the codes and will present the quantitative results of them if time permits before the end of the discussion. In addition, we already added the visualized results of them on RealEstate10K on the supplimentary materials (i.e. mv\_splat.mp4 and pixelSplat\_scene\_result.jpg), which also indicated their problems (including view inconsistency).

---

> > > > > > > ### Author Response · Authors · 2024-12-03
> > > > > > >
> > > > > > > Dear Reviewer t736,
> > > > > > >
> > > > > > > Thank you for dedicating your time and providing feedback on our work. We have modified our paper according your new suggestions and will update them in the camera-ready version if the paper is accepted. The experiment of training MVSplat on Objaverse dataset is still running and we will present the results as soon as possible  (within the discussion period) when it convergent. As it approaches the end of the discussion period, we really want to know do you still have other concerns or questions so that we can put efforts in the last day to solve them. Your reply is very important for us, and we are looking forward to it.

---

> > > > > > > > ### Author Response · Authors · 2024-12-04
> > > > > > > >
> > > > > > > > We trained the MVSplat on the Objaverse dataset and tested it on the GSO dataset (consistent with the setting of ours). After around 100,000 iterations for training, it appeals to have converged (with little loss decrease and oscillating PSNR on the validation set). The quantitative results on the GSO dataset are shown in the below table, where it have inferior performance than ours no matter whether masking the 3D Gaussians corresponding the background pixels. We also visualized some results of 3D Gaussians centers and still observed obvious misaligned 3D Gaussians from different views (view inconsistency). We plan to update these results in Appendix A.5 and replace the Figure 21 and Table 12 with the new results. As we have modified our paper and added the new experiments following your suggestions, we sincerely hope you can reconsider your current rating on our work since your major concerns have been addressed and you also felt our work is on the bar for acceptance as you said in previous replies.
> > > > > > > >
> > > > > > > > ## Comparison with MVSplat on the GSO-random dataset in the 4-view input setting
> > > > > > > >
> > > > > > > > | Method             | PSNR ↑   | SSIM ↑ | LPIPS ↓ | Rendering time |
> > > > > > > > |--------------------|----------|--------|---------|----------------|
> > > > > > > > | MVSplat            | 23.06    | 0.90   | 0.13    |  0.0090         |
> > > > > > > > | MVSplat (masked)   | 24.10    | 0.91   | 0.12    | 0.0045         |
> > > > > > > > | Ours               | **26.30**| **0.93** | **0.08** | **0.0019**    |

---

> ### Author Response · Authors · 2024-11-28
>
> Similar to LGM, both pixelSplat [1] and MVSplat [2] follow a workflow that regress Gaussians from each view within the respective camera spaces and subsequently merge them in the world space. In pixelSplat, the integration of cross-view-aware features is through an epipolar Transformer, and it still suffers from inaccurate depth estimation. MVSplat adopts a design that incorporates a cost volume storing cross-view feature similarities for all possible depth and makes a more accurate depth prediction. However, they assign each pixel with a 3D Gaussian and thereby generates a planar representation rather than the object itself. In addition, MVSplat tends to obscure object details due to the occlusion by 3D Gaussians from other viewpoints, resulting in suboptimal outcomes. To address this issue, we mask the 3D Gaussians on background pixels to help it focus on rendering 3D Gaussians contributing to the object itself, noted as 'MVSplat (masked)' in the results.
>
> We present the comparison to pixelSplat [1] and MVSplat [2] in Appendix A.5 and the quantitative results on the GSO-random dataset is shown in Table 12. From the table, we can see that their results is significantly worse than ours. It is probably due to the fact that they have only been trained on the scene reconstruction dataset RealEstate10 [3], which only contains small camera difference among views. The cameras of object reconstruction dataset GSO-random has larger variations, so we observe more severe misaligned 3D Gaussians (view inconsistency) from different input views for MVSplat, as shown in the visualized results in Figure 21 (we also add the corresponding videos and ply files in the MVSplat\_results folder of supplementary materials). And we find that MVSplat cannot correctly predict the back side of the object. It is not a big issue for scene reconstruction as their camera only moves a little, but would lead to incomplete reconstruction of objects. In the figure, we present the centers of 3D Gaussians generated from different views with different colors and the novel views are rendered from the 3D Gaussians from all views. As for pixelSplat, it almost cannot output reasonable results when use GSO-random dataset for testing, so we have not presented their visualized results. We provide the content of Table 12 as following:
>
> **Table: Comparison with MVSplat and pixelSplat on the GSO-random dataset**
>
> | Method                    | PSNR ↑  | SSIM ↑ | LPIPS ↓ | Inference time ↓ | Rendering time ↓ |
> |---------------------------|--------|--------|---------|----------------|----------------|
> | MVSplat                   | 12.92  | 0.80   | 0.30    | 0.112          | 0.0090         |
> | MVSplat (masked)          | 16.52  | 0.80   | 0.19    | 0.112          | 0.0045         |
> | pixelSplat (2 views)      | 12.00  | 0.80   | 0.28    | 1.088          | 0.0045         |
> | pixelSplat (2 views masked)| 12.05 | 0.79   | 0.27    | 1.088          | 0.0023         |
> | **Ours**                  | **26.30** | **0.93** | **0.08** | **0.694**   | **0.0019**     |
>
> [1] David Charatan, et al., pixelSplat: 3D Gaussian Splats from Image Pairs for Scalable Generalizable 3D Reconstruction. CVPR2024.
>
> [2] Chen, MVSplat: Efficient 3D Gaussian Splatting from Sparse Multi-View Images. ECCV2025
>
> [3] Zhou, et al., Stereo magnification: Learning view synthesis using multiplane images. ACM Trans. Graph. (Proc.
> SIGGRAPH), 2018

---

> > ### Author Response · Authors · 2024-12-02
> >
> > Dear Reviewer t736,
> >
> > Thank you for dedicating your time and providing feedback on our work. We have tried our best to address the concerns you previously raised by providing additional explanations or conducting further experiments, and we have rectified writing issues.
> > We kindly seek your thoughtful reconsideration for a potential score increase, taking into account of the revisions made based on your invaluable feedback if you have no further concerns. Your time and insights are sincerely valued. If you have further questions, we are also pleased to answer you in the rest discussion period.

---

### Official Review · Reviewer_UhUJ · 2024-11-03

**Soundness:** 3
**Presentation:** 2
**Contribution:** 3
**Rating:** 8
**Confidence:** 3

**Summary:**

The paper proposes a method for 3D object reconstruction (with 3D Gaussians) by employing a novel encoder-decoder framework. The method operates in a two-stage process: (a) A coarse initialization provides initial 3D Gaussians leveraging only a small subset of the input images. (b) In the refinement stage, the Gaussians are optimized using multi-view deformable attention and spatially efficient self-modules. The final 3D Gaussians are obtained by updating initial estimates based on the multi-view features.

**Strengths:**

- The experimental results seem particularly strong with large improvements over the baselines.

**Weaknesses:**

- The introduction and the first part of the paper, in general, are badly written. It gives the feeling that it was blindly polished by an LLM that had no idea about the topic. There are many unexplained concepts or ones that seemingly mean nothing. It uses fancy words and expressions that ultimately mean nothing and only make it harder to understand what is going on in the paper. I detail this in the minor comments section.
- The title is misleading. What the authors do is 3D Object Reconstruction and not general 3D Reconstruction. I also have a hard time accepting 3DGS as a reconstruction method since, to me, reconstruction would involve estimating the camera parameters as well (which is considered given here). However, this is only my concern, and I am fine if the authors go with "3D Object Reconstruction".
- Missing comparison to other sparse-view methods, e.g., MVSplat and pixelSplat. The authors propose a sparse-view pipeline so it would be fair to compare with other similar methods. I know that MVSplat and pixelSplat reconstruct the entire scene while the proposed method only has an object. However, they should still be compared as I see no fundamental limitation that would prevent MVSplat/pixelSplat from being applied here.

The most important comment here from my side is: (a) There are missing comparisons that should be added. (b) The introduction and beginning of the paper should be rewritten. (c) The title and narrative should be changed a bit.

Minor comments:
- L040 "view inconsistency" I am a bit unsure what this means here. I suggest the authors explain clearly here what this issue is as they build the rest of their introduction on this.
- L044 "view-specific camera space" What is a view-specific camera space?
- L045 "These Gaussians are then converted to world space". This sentence makes no sense.
- L049 This paragraph is full of abbreviations without any explanation for them. The authors state that they use a DETR-like Transformer and they are inspired by DIG3D and TAPTR, but this says nothing without having to read all these papers. This is not a good way of writing an introduction. The authors should make sure that the general concepts and ideas are understandable just from reading the text they provide.
- L070 "More specifically, MVDFA utilize camera modulation techniques (Karras et al., 2019; Hong et al., 2024) to diversify queries based on views." Similary as before, I have no idea what camera modulation techniques are, the authors don't even provide a single example. I had to open the cited papers and read about it, which really does break the flow of reading the introduction.
- L074 "our model prioritizes multi-view distinctions to achieve a more precise 3D representation." - Again, what does "multi-view distinction" mean? This entire thing sounds like it was polished by an LLM that had no idea what really happens.
- L140 "In total" This is not needed.
- L151 The authors say that they use a "unitary 3D Gaussians representation" but they never explain what such a representation is. Also, typo: "3D Gaussians representation" -> "3D Gaussian representation"
- L158 "Spatially efficient self-attention" -> What is a spatially efficient self-attention?

**Questions:**

See in the weaknesses section.

---

> ### Author Response · Authors · 2024-11-20
>
> Thank you very much for your thoughtful comments on our work. We greatly appreciate your feedback. We will address each of your concerns individually as outlined below. For those that require additional experiments, we will ensure to upload the results as soon as possible. Should you have any further questions or concerns, please do not hesitate to reach out to us.
>
> # For Weaknesses1:
> Thanks a lot for your so detailed writting comment.
> Sorry for causing the bad reading experience, we explained the confusing sentences in the following and revised our paper to reduce unclear expressions.
>
> ## L040 "view inconsistency":
> View inconsistency means that 3D reconstructions from various input views are misaligned because of the inaccurate depth prediction from single view separately, which can be clearly illustrated by the attached video (inconsistentpc.mp4). Another example is that, in Figure 1, the handle of the pot generated from input image 1 appears at a different position compared to the handle from input image 2. This difference arises also due to inaccurate depth predictions of each view, leading to spatial variations and resulting in multiple handles being rendered in the views. We have added this explanation in our paper.
>
> ## L044 view-specific camera space:
> In the corresponding sentence, we want to point out that, for methods like Splatter Image [1] and LGM [3], the 3D Gaussians for each view are first reconstructed in the camera space of the corresponding input view, then they are merged in the world space after the camera-to-world space transformation. Therefore, in this context, "view-specific" is to stress that they predict 3D Gaussians in a view-independent manner. We have modified this statement in the revised version.
>
> ## L045 "These Gaussians are then converted to world space":
> For methods Splatter Image [1] and LGM [2], the 3D Gaussians are first reconstructed in the camera space of each input view, then they are merged in the world space after the camera-to-world space transformation. Therefore, the step of transforming 3D Gaussians from the camera space to the world space cannot be ignored, and it is the main cause of the view inconsistent problem.
>
> ## L049 The authors should make sure that the general concepts and ideas are understandable just from reading the text they provide:
> Thank you for your suggestions, we will modify these in the revised version. DETR-like models link object bounding box as queries and treat image tokens as keys and values in Transformer, which have made a great success. We borrow the similar philosophy that links each 3D Gaussian with queries and also treat image tokens as keys and values, then refine 3D Gaussians iteratively. % We will use more widely accepted paper and explain DETR briefly.
>
> ## L070 camera modulation:
> Sorry for the bad reading experience again, we have modified the paper following your suggestions. To be clear, camera modulation means we linearly transform image features of each view, say $F' = WF + b$, where $F$ is the original image feature, $W$ and $b$ are weights and bias regressed by an MLP with camera parameters as input. Such operation gives each view its corresponding camera pose information. We have added a brief explanation in the introduction.
>
> ## L074 "multi-view distinctions:
> The "multi-view distinction" means that we use camera modulation to distinct queries before projecting them to each view to retrieve image features, so as that queries can be aware of different camera poses.
>
> ## L140 "In total" This is not needed:
> Thanks for pointing it out, we will delete it.
>
> ## L151 unitary 3D Gaussians representation: Also, typo: "3D Gaussians representation" -> "3D Gaussian representation".
>
> Unitary 3D Gaussian representation means we define a unique set of 3D Gaussians in the world space no matter how many input views are given. By contrast, previous methods predict one set of 3D Gaussians in camera space for each input view, so there will be multiple sets of 3D Gaussians given multiple input views, and then merge them together in the world space to get the final output 3D Gaussians. We will fix the typo.
>
> ## L158 "Spatially efficient self-attention":
> Thanks for pointing this out, we miss the reference to Section 3.2.3 and will add it in the revised version.
> Spatially efficient self-attention means to do self-attention in a memory efficient way by sampling part of 3D Gaussians as keys and values, and we detailed this part in the Section 3.2.3.
>
> [1] Stanislaw Szymanowicz, et al., Splatter Image: Ultra-Fast Single-View 3D Reconstruction.CVPR2024.
>
> [2] Zi-Xin Zou, et al., Triplane Meets Gaussian Splatting: Fast and Generalizable Single-View 3D Reconstruction with Transformers. CVPR2024.
>
> [3] Jiaxiang Tang, et al., LGM: Large Multi-View Gaussian Model for High-Resolution 3D Content Creation. ECCV2024.

---

> ### Author Response · Authors · 2024-11-20
>
> # For Weaknesses2:
> We agree that "3D recontruction" in earlier literature usually include estimating camera parameters, but here we use "3D reconstruction" to denote the reconstruction of 3D Gaussians following recent 3D GS papers Splatter Image[1], Triplane-Gaussian[2]. We also discussed the camera pose problem in the limitation part in our paper.
>
> # For Weaknesses3:
> Thank you for pointing this out.
> We will add the comparison as soon as possible.
>
> [1] Stanislaw Szymanowicz, et al., Splatter Image: Ultra-Fast Single-View 3D Reconstruction.CVPR2024.
>
> [2] Zi-Xin Zou, et al., Triplane Meets Gaussian Splatting: Fast and Generalizable Single-View 3D Reconstruction with Transformers. CVPR2024.
>
> [3] Jiaxiang Tang, et al., LGM: Large Multi-View Gaussian Model for High-Resolution 3D Content Creation. ECCV2024.

---

> > ### Comment · Reviewer_UhUJ · 2024-11-25
> >
> > Thank you for the answers. Should I expect results from MVSplat or pixelSplat within the rebuttal period?

---

> > > ### Author Response · Authors · 2024-11-25
> > >
> > > Thank you for your reply, we think we can provide the results from MVSplat or pixelSplat within the rebuttal period. We are trying to test them on the GSO dataset to compare with our method. As they are initially designed for scene reconstruction, the datasets they used (RealEstate10K [1] and ACID [2]) has different camera system convention with the GSO dataset, so it may take some time to align them. We have successfully run their official codes on the dataset of RealEstate10K and visualized the centers of 3D Gaussians in each view, where we also observe the view inconsistent problem. The results are shown in the supplementary video mv\_splat.mp4.
> > >
> > > [1] Zhou, et al., Stereo magnification: Learning view synthesis using multiplane images. ACM Trans. Graph. (Proc.
> > > SIGGRAPH), 2018
> > >
> > > [2] Liu, et al., Infinite nature:
> > > Perpetual view generation of natural scenes from a single image. ICCV 2021.

---

> > > ### Author Response · Authors · 2024-11-28
> > >
> > > Similar to LGM, both pixelSplat [1] and MVSplat [2] follow a workflow that regress Gaussians from each view within the respective camera spaces and subsequently merge them in the world space. In pixelSplat, the integration of cross-view-aware features is through an epipolar Transformer, and it still suffers from inaccurate depth estimation. MVSplat adopts a design that incorporates a cost volume storing cross-view feature similarities for all possible depth and makes a more accurate depth prediction. However, they assign each pixel with a 3D Gaussian and thereby generates a planar representation rather than the object itself. In addition, MVSplat tends to obscure object details due to the occlusion by 3D Gaussians from other viewpoints, resulting in suboptimal outcomes. To address this issue, we mask the 3D Gaussians on background pixels to help it focus on rendering 3D Gaussians contributing to the object itself, noted as 'MVSplat (masked)' in the results.
> > >
> > > We present the comparison to pixelSplat [1] and MVSplat [2] in Appendix A.5 and the quantitative results on the GSO-random dataset is shown in Table 12. From the table, we can see that their results is significantly worse than ours. It is probably due to the fact that they have only been trained on the scene reconstruction dataset RealEstate10 [3], which only contains small camera difference among views. The cameras of object reconstruction dataset GSO-random has larger variations, so we observe more severe misaligned 3D Gaussians (view inconsistency) from different input views for MVSplat, as shown in the visualized results in Figure 21 (we also add the corresponding videos and ply files in the MVSplat\_results folder of supplementary materials). And we find that MVSplat cannot correctly predict the back side of the object. It is not a big issue for scene reconstruction as their camera only moves a little, but would lead to incomplete reconstruction of objects. In the figure, we present the centers of 3D Gaussians generated from different views with different colors and the novel views are rendered from the 3D Gaussians from all views. As for pixelSplat, it almost cannot output reasonable results when use GSO-random dataset for testing, so we have not presented their visualized results. We provide the content of Table 12 as following:
> > >
> > > **Table: Comparison with MVSplat and pixelSplat on the GSO-random dataset**
> > >
> > > | Method                    | PSNR ↑  | SSIM ↑ | LPIPS ↓ | Inference time ↓ | Rendering time ↓ |
> > > |---------------------------|--------|--------|---------|----------------|----------------|
> > > | MVSplat                   | 12.92  | 0.80   | 0.30    | 0.112          | 0.0090         |
> > > | MVSplat (masked)          | 16.52  | 0.80   | 0.19    | 0.112          | 0.0045         |
> > > | pixelSplat (2 views)      | 12.00  | 0.80   | 0.28    | 1.088          | 0.0045         |
> > > | pixelSplat (2 views masked)| 12.05 | 0.79   | 0.27    | 1.088          | 0.0023         |
> > > | **Ours**                  | **26.30** | **0.93** | **0.08** | **0.694**   | **0.0019**     |
> > >
> > > [1] David Charatan, et al., pixelSplat: 3D Gaussian Splats from Image Pairs for Scalable Generalizable 3D Reconstruction. CVPR2024.
> > >
> > > [2] Chen, MVSplat: Efficient 3D Gaussian Splatting from Sparse Multi-View Images. ECCV2025
> > >
> > > [3] Zhou, et al., Stereo magnification: Learning view synthesis using multiplane images. ACM Trans. Graph. (Proc.
> > > SIGGRAPH), 2018

---

> > > > ### Author Response · Authors · 2024-12-02
> > > >
> > > > Dear Reviewer UhUJ,
> > > >
> > > > Thank you for dedicating your time and providing feedback on our work. We have tried our best to address the concerns you previously raised by providing additional explanations or conducting further experiments, and we have rectified writing issues.
> > > > We kindly seek your thoughtful reconsideration for a potential score increase, taking into account of the revisions made based on your invaluable feedback if you have no further concerns. Your time and insights are sincerely valued. If you have further questions, we are also pleased to answer you in the rest discussion period.

---

> > > > > ### Comment · Reviewer_UhUJ · 2024-12-02
> > > > >
> > > > > Thanks for the answers and additional experiments. I am happy with the paper.

---

> > > > > > ### Author Response · Authors · 2024-12-02
> > > > > >
> > > > > > Thank you agian for your positive feedback and kind support!
> > > > > > We sincerely appreciate your recognition of our efforts and contribution.

---

### Official Review · Reviewer_9662 · 2024-11-03

**Soundness:** 3
**Presentation:** 3
**Contribution:** 3
**Rating:** 5
**Confidence:** 4

**Summary:**

This paper introduces UniG, a novel 3D reconstruction and novel view synthesis model that creates high-fidelity 3D Gaussian Splatting from sparse images while maintaining view consistency. To tackle the view inconsistency issue  in traditional 3D Gaussian-based methods which directly regressing Gaussians per-pixel for each view, the authors proposed to employ a DETR-like framework that uses 3D Gaussians as decoder queries, refining their parameters through multi-view cross-attention (MVDFA) across input images. This design allows for an arbitrary number of input images without causing a memory explosion, as the number of 3D Gaussians used as queries is independent of the input views. Comprehensive experiments demonstrate UniG's superiority over existing methods in terms of quantitatively and qualitatively.

**Strengths:**

1. he pipeline in this submission is technically sound and is clearly written and well organized.

2. The authors, drawing inspiration from DETR, propose a Gaussian-based 3D reconstruction and novel view synthesis approach which can achieve SOTA performance. Extensive experiments have validated the model's effectiveness and outstanding performance.

3. For the comparison, the numerical results show a significant performance improvement over the baseline method in GSO data. And for the ablation study, the authors show the importance of some designs like the coarse stage initialization and refinement.

**Weaknesses:**

1. Overall, the structure of this paper resembles a multi-view version of a combination between TriplaneGaussian and Instant3D and the importance of the MVDFA module and the two stages is not very convincing.

2. Although the authors have proposed the MVDFA module to integrate coarse and refine information, attempting to address the inconsistency issue of LGM when predicting from each perspective. However, aside from Figure 1, there are no more images demonstrating the severity of this inconsistency. Additionally, what would be the result by using a simple mask on the LGM or Splatter Image prediction.

3. A minor thing is that in Table 2, Splatter Image appears to show promising performance, similar to the results of the coarse stage proposed in the paper, but there is a lack of visual comparison with it.

[1] Zi-Xin Zou, et al., Triplane Meets Gaussian Splatting: Fast and Generalizable Single-View 3D Reconstruction with Transformers, CVPR 2024

[2] Jiahao Li, et al., Instant3D: Fast Text-to-3D with Sparse-View Generation and Large Reconstruction Model, ICLR 2024

**Questions:**

1. In Figure 2, why does the feature extractor of the coarse network output 3D GS, while the same module in the refinement network outputs multiview feature maps?

2. In line 152, the paper states that "during the coarse stage, one or more images are randomly selected." This raises questions about whether a frontal view image is necessary at the coarse stage, or if an arbitrary view would suffice? If an arbitrary view is acceptable, how is the correctness of the coarse output ensured? Furthermore, the paper lacks an ablation study on the number of images used during the coarse stage.

3. In the refinement stage, the 3D GS output by the coarse network are used as input for the MVDFA module. However, since the coarse model only uses a single view image as input, the 3D GS generated during the coarse stage may not align with the structure of the four-view input in the refinement stage. This raises the question of whether this discrepancy could impact the MVDFA model? In other words, how can we address the consistency issue between the 3D GS from the coarse stage and the multi-view images in the refinement stage?

**Details Of Ethics Concerns:**

1. In Figure 2, why does the feature extractor of the coarse network output 3D GS, while the same module in the refinement network outputs multiview feature maps?

2. In line 152, the paper states that "during the coarse stage, one or more images are randomly selected." This raises questions about whether a frontal view image is necessary at the coarse stage, or if an arbitrary view would suffice? If an arbitrary view is acceptable, how is the correctness of the coarse output ensured? Furthermore, the paper lacks an ablation study on the number of images used during the coarse stage.

3. In the refinement stage, the 3D GS output by the coarse network are used as input for the MVDFA module. However, since the coarse model only uses a single view image as input, the 3D GS generated during the coarse stage may not align with the structure of the four-view input in the refinement stage. This raises the question of whether this discrepancy could impact the MVDFA model? In other words, how can we address the consistency issue between the 3D GS from the coarse stage and the multi-view images in the refinement stage?

---

> ### Author Response · Authors · 2024-11-20
>
> Thank you very much for your thoughtful comments on our work. We greatly appreciate your feedback. We will address each of your concerns individually as outlined below. For those that require additional experiments, we will ensure to upload the results as soon as possible. Should you have any further questions or concerns, please do not hesitate to reach out to us.
>
> # For Weaknesses1:
> Our method differs from Triplane-Gaussian in the following aspects. First, Triplane-Gaussian is a single-view reconstruction method so it does not need to consider the multi-view information fusion problem while our method target at reconstructing a unitary set of 3D Gaussians with arbitrary number of input views, resolving the view inconsistent problem that only happens in the multi-view input setting. Second, Triplane-Gaussian adopt the triplane representation, which would lead to a lack of detailed information in the 3D structure and imposing a rigid grid alignment that limits flexibility (LGM[1], PointNet[2]). Third, Triplane-Gaussian requires 3D supervision to achieve a good performance while our method only requires multi-view 2D images. Even so, our method still outperforms the Triplane-Gaussian given the single-view image as input, as shown in Table 5 in the revised version.  We also provide the table here.
>
> **Table: Quantitative results trained on Objaverse LVIS and tested on GSO. 3D sup. means need 3D supervision.**
>
> | Method                           | PSNR ↑   | SSIM ↑ | LPIPS ↓ | 3D sup. | Inference time |
> |----------------------------------|----------|--------|---------|---------|----------------|
> | Triplane-Gaussian             | 18.61    | 0.853  | 0.159   | ✔       | 1.906          |
> | Ours                             | **23.45**| **0.897** | **0.093** | ✘       | **0.476**     |
>
> As for Instant3D, it also utilize the triplane and apply the NeRF to represent the reconstructed object. The NeRF representation is fundamentally different from 3D Gaussians and cannot be naively replaced. Therefore, our method is essentially different from the mentioned methods. MVDFA is designed to to save memory occupation and training and inference time so that we can set a larger number of 3D Gaussians to represent an object. The two-stage framework is also important because we empirically found that a good initialization for 3D Gaussians would make a big difference, as shown in Table 4 and Table 10 in our paper.
>
> # For Weaknesses2:
> As shown in the Figure 7 in the revised PDF, we visualize the predicted 3D Gaussians centers of LGM and Splatter Image and paint them of different views with different colors. From the figure, the misalignment of 3D Gaussians from different views can be obviously seen, which is what we call "view inconsistency", while our method share a unitary set of 3D Gaussians for each view so there is no such severe view inconsistent problem. Moreover, in Figure 4, we also shows the view inconsistency example of LGM that predict not aligned objects from different views.
>
> When masks are applied to remove backgrounds, as shown in Figure 16 and the attached vedio (masked\_point\_cloud.mp4), the misaligned 3D Gaussians can be alleviated to some extent but the problem still exists. The corresponding quantitative results are presented in Table 6. Removing background points lead to less outliers and better rendering results but there still are obvious artifacts that can be observed. The Table is also given here:
>
> **Table: Comparison between masked and original pixel-aligned methods**
>
> | Method                      | PSNR ↑ | SSIM ↑ | LPIPS ↓ |
> |-----------------------------|--------|--------|---------|
> | LGM                         | 17.4810| 0.7829 | 0.2180  |
> | LGM (masked)                | 21.6008| 0.8608 | 0.1232  |
> | Splatter Image              | 25.6241| 0.9151 | 0.1517  |
> | Splatter Image (masked)     | 25.0648| 0.9147 | 0.1684  |
>
> # For Weaknesses3:
> The coarse stage share a similar structure with the previous method (Splatter Image [3]) whose function is to provide a coarse initialization for the refinement stage to avoid hard convergence problem. The results in Table 2 and Table 4 use different validation dataset, so their numerical values cannot be directly compared. The visualized comparison of Splatter Image and the coarse stage of our model are shown in Figure 15. From the figure, we can see that the output of the coarse stage is not as good as Splatter Image, but with the refinement stage, the final output outperforms the Splatter Image.
>
> [1] Jiaxiang Tang, et al., LGM: Large Multi-View Gaussian Model for High-Resolution 3D Content Creation. ECCV2024.
>
> [2] Charles R Qi, et al., PointNet: Deep Learning on Point Sets for 3D Classification and Segmentation. CVPR2017a.
>
> [3] Stanislaw Szymanowicz, et al., Splatter Image: Ultra-Fast Single-View 3D Reconstruction. CVPR2024.

---

> ### Author Response · Authors · 2024-11-20
>
> # For Questions1:
> Thank you for pointing it out. Actually, the feature extractor for both stages will output feature maps first, then for the coarse stage, a convolution layer (omitted in the figure) is used to regress pixel-aligned 3D Gaussians as coarse initialization. We have modified the figure to make it clear in the revised PDF.
>
> # For Questions2:
> The function of coarse initialization is mainly to avoid out-of-boundary projected points, thus minor variations will not make a big difference for the refinement stage. In the coarse stage, no matter which view is selected, the 3D Gaussians are first reconstructed in its camera space, and then transformed to the world space (the camera space of the first view) using camera pose parameters. Therefore, the concept of "front view" does not exist in this context. No matter which view is selected to be the input of the coarse stage, it will not make a big difference. We also add the ablation study on the number of images used during the coarse stage. As shown in Table 9 of the revised PDF, the number of images used during the coarse stage does not influence the final result. We also show the content of Table 9 here.
>
> **Table: Ablation study results of different views and different numbers of views for the coarse stage (with 4 views in the refinement stage)**
>
> | Number of views in coarse stage | PSNR ↑ | SSIM ↑ | LPIPS ↓ |
> |---------------------------------|--------|--------|---------|
> | 1                               | 30.2312| 0.9608 | 0.0413  |
> | 2                               | 30.4245| 0.9614 | 0.0422  |
> | 3                               | 30.3442| 0.9618 | 0.0419  |
> | 4                               | 30.4521| 0.9620 | 0.0412  |
>
> # For Questions3:
> The input images (1 or 2 views) for the coarse stage are sampled from the all multi-view input images, and the 3D Gaussians are first reconstructed in the camera space and then transformed to the world space (the camera space of the first input view). The 3D Gaussians in the same world space will be the initialization of the refinement stage, which are already coarsely aligned with the other input views. Their final positions and other parameters will be iteratively refined to align all input views in the refinement stage.
>
> [1] Jiaxiang Tang, et al., LGM: Large Multi-View Gaussian Model for High-Resolution 3D Content Creation. ECCV2024.
>
> [2] Charles R Qi, et al., PointNet: Deep Learning on Point Sets for 3D Classification and Segmentation. CVPR2017a.
>
> [3] Stanislaw Szymanowicz, et al., Splatter Image: Ultra-Fast Single-View 3D Reconstruction. CVPR2024.

---

> ### Author Response · Authors · 2024-11-25
>
> Dear Reviewer 9662,
>
> We want to express our sincere gratitude for your insightful suggestions which are instrumental in enhancing the quality of our work. We hope that our proposed modifications would have addressed your concerns about the clarity of our presentation. We would really appreciate it if you could let us know if there are any further questions or aspects of the paper that require additional clarification. Thank you once again for your time and consideration.

---

> > ### Comment · Reviewer_9662 · 2024-11-29
> >
> > I have read all the rebuttals and am very grateful for your responses and the additional experimental results.
> >
> > I can understand the differences between this work and TriplaneGaussian and Instant3D. However, like Reviewer t736, as I mentioned in Weakness 2, there needs to be more elaboration and explanation regarding the severity of the inconsistency caused by per-pixel Gaussian prediction; it is not currently demonstrated that this issue is widespread and severe.
> >
> > In addition, I still do not understand how the consistency between the coarse stage's GS and the refinement stage's GS can be guaranteed. The authors only said that the same structure was used in the coarse stage, but how can it be guaranteed that the results of the single-view prediction in the coarse stage will be consistent with the multi-view images input in the refinement stage? Or perhaps I have misunderstood something.

---

> > > ### Author Response · Authors · 2024-11-30
> > >
> > > Thank you for your reply. We offer additional clarification addressing your concerns as follows:
> > >
> > > # For the concern about more elaboration and explanation regarding the severity of the inconsistency
> > > To present the severity of the inconsistency caused by per-pixel Gaussian prediction, we have added more visualized results in the supplementary materials (Figure 16, Figure 21). We can observe that 3D Gaussians from different views (in different colors) are misaligned. On the other hand, we have tested most of the high impact open-sourced per-pixel methods (including LGM, Splatter Image, PixelSplat, MVSplat) on the GSO datasets and visualized their 3D Gaussian centers in different colors for each view. As shown in the video inconsistentpc.mp4, masked\_point\_cloud.mp4, and the videos in MVSplat\_results folder, it is evident that the Gaussians depicted from different views in different colors do not align when representing the same object segment. This misalignment leads to blurred final rendering outcomes and can even produce 'ghosting' artifacts, as demonstrated by the presence of two shoes in the rendered novel views in Figure 16. This inconsistency, characterized by such 'ghosting' artifacts, is widespread and notably observed in Figures 4 and 21 as well. As for GRM and GS-LRM, we have not found obvious problems from the results in their websites, but we think they should be considered more as concurrent works and they have not released their codes for fair comparison. Therefore, we may generally conclude that almost all existing per-pixel methods have the inconsistency issue.
> > >
> > > # For the concern about the consistency between the coarse stage and the refinement stage
> > > For the concern about the consistency between the coarse stage and the refinement stage, we use an example to illustrate. For example, we have 4 input views, named $V_1, V_2, V_3, V_4$ and their corresponding camera parameters $\pi_1, \pi_2, \pi_3, \pi_4$. Without loss of generality, we use $V_1$ as the input of the coarse stage and use its camera space as the world space. We then transform all camera parameters to the world space and get the new camera parameters $\pi_{1}', \pi_{2}', \pi_{3}', \pi_{4}'$. Here, $\pi_{1}'$ is the identity matrix because we define the camera space of $V_1$ as world space. Then, we generate $N$ 3D Gaussians $G_{init}$ from $V_1$ under the world space from the coarse stage and $G_{init}$ as the initialization of our refinement stage. After that, we project $G_{init}$ onto all the 4 views with the new camera parameters  $\pi_{1}', \pi_{2}', \pi_{3}', \pi_{4}'$ and gather the information to update $G_{init}$ layer by layer. Throughout the entire process, a single set of 3D Gaussians is defined in world space, utilized in both the coarse and refinement stages. All the input views contribute to this singular set of 3D Gaussians in both stages. Consequently, since the same set of 3D Gaussians is maintained across both the coarse and refinement stages, i.e., we do not have separate predictions for the coarse stage and refinement stage, the inherent consistency of this shared representation precludes the possibility of introducing any inconsistencies.

---

> > > > ### Author Response · Authors · 2024-12-02
> > > >
> > > > Dear Reviewer 9662,
> > > >
> > > > Thank you for dedicating your time and providing feedback on our work. We have tried our best to address the concerns you previously raised by providing additional explanations or conducting further experiments, and we have rectified writing issues.
> > > > We kindly seek your thoughtful reconsideration for a potential score increase, taking into account of the revisions made based on your invaluable feedback if you have no further concerns. Your time and insights are sincerely valued. If you have further questions, we are also pleased to answer you in the rest discussion period.

---

> > > > > ### Author Response · Authors · 2024-12-03
> > > > >
> > > > > Dear Reviewer 9662,
> > > > >
> > > > > Thank you for dedicating your time and providing feedback on our work. We have presented new experiments and more explanations to address your concerns or questions. As it approaches the end of the discussion period, we really want to know do you still have other concerns or questions so that we can put efforts in the last day to solve them. We do not wish for you to have negative attitude to our paper due to any misunderstanding. Your reply is very important for us, and we are looking forward to it.

---

### Official Review · Reviewer_4FKX · 2024-11-04

**Soundness:** 3
**Presentation:** 2
**Contribution:** 3
**Rating:** 6
**Confidence:** 4

**Summary:**

The paper introduces UniG, a new 3D reconstruction and novel view synthesis model leveraging unitary 3D Gaussians for view-consistent 3D scene representation from sparse posed image inputs. Existing 3D Gaussians-based methods usually regress per-pixel 3D Gaussian for each view, create 3D Gaussians per view separately, and merge them through point concatenation. Such a view-independent reconstruction, which often results in a view inconsistency issue. UniG addresses view inconsistency in existing methods by DETR (DEtection TRansformer)-like framework, treating 3D Gaussians as decoder queries updated layer by layer
by performing multi-view cross-attention over multiple input images. This design allows UniG to maintain a single 3D Gaussian set, supporting arbitrary input views without memory expansion and ensuring consistency across views.
Experiments validate UniG's superior performance in 3D reconstruction on Objaverse and GSO datasets, achieving better results than the selected baselines qualitatively and quantitatively.

**Strengths:**

1. The motivation is sound and clear.
2. The proposed methods demonstrate improved performance in 3D reconstruction on the Objaverse and GSO datasets, achieving better results both qualitatively and quantitatively compared to selected baselines.
3.  The proposed methods exhibit scalability with arbitrary views: despite being trained on a fixed number of views, UniG can handle an arbitrary number of input views without a significant increase in memory usage.
4. The paper presents adequate experiments and provides a thorough ablation of the design choices in the methods.

**Weaknesses:**

1. The visual results are blurry and have obvious artifacts. The resolution is low (no larger than 512)
2. Under some cases, the improvements are not obvious compared with previous methods, as shown in Table 2, the PNSR improvement is only ~0.5dB.

**Questions:**

1. In Fig. 5, the authors demonstrate that PSNR performance improves as the number of input views increases (from 2 to 8). I am curious about the effect of continuing to increase the number of input views beyond 8. Will performance decline after reaching this point? We can observe that the performance gain diminishes when the input view count increases from 6 to 8, suggesting a potential decline in performance if this number exceeds 8.

2. In line 374, the authors state that "previous methods rely on fixed views as input," which leads to a performance drop when random input views are used. By comparing Tables 1 and 2, it appears that this method also experiences a notable performance decline (a reduction of approximately 4 dB in PSNR) with random inputs. Interestingly, however, the baseline method Splatter Image does not show this performance drop (its PSNR increases slightly from 25.6 to 25.8). This suggests that Splatter Image demonstrates superior generalization regarding input view pose distribution compared to this method. I am interested in the authors’ explanation for this difference.

---

> ### Author Response · Authors · 2024-11-20
>
> Thank you very much for your thoughtful comments on our work. We greatly appreciate your feedback. We will address each of your concerns individually as outlined below. For those that require additional experiments, we will ensure to upload the results as soon as possible. Should you have any further questions or concerns, please do not hesitate to reach out to us.
>
> # For Weaknesses1:
> Thanks for pointing this out. The blurry are mainly caused by the automatic zoom-in of the PDF editing software as our model is trained with the resolution of 128, so as the rendered output image. Now, we are working on the result with the resolution of 512 to solve this problem.
>
> # For Weaknesses2:
> Apart from PSNR, other metrics SSIM $\uparrow$ (increased by 0.3) and LPIPS $\downarrow$ (reduced by 0.1739) are also significantly improved. We provide the visualization on Splatter Image and our method in Figure 15 in the revised version. Although the improvement for PSNR is not significant, the visualization of our model is much better than Splatter Image. (The visualizations for other methods are in Figure 10).
> Moreover, previous novel view synthesis papers like MVGamba[1], Splatter Image[2], Instantmesh[3], pixelsplat[4] also provide PSNR $\uparrow$ improvement around 0.5dB, so it is not a marginal increase.
>
> # For Questions1:
> We give more view result in Appendix Figure 17 in the revised PDF.
> Our model is positioned on the 'sparse view' setting, which indicates the number of views less then 10, so we only reports the performance of views from 2 to 8. With the increase of input views, information from similar views becomes redundant, so the gain for our model has become plateaued while other methods suffer from performance drop as they cannot handle too many input views due to the view inconsistent problem. As we keep increasing the number of input views larger than 8, our method can still benefit from more input views (as shown in Appendix Figure 17) while others meet the CUDA-out-of-memory problem.
>
> # For Question2:
> Thanks a lot for pointing this out. The setting of random input view is obvious a more challenging task than the setting of fixed input view, thus our method also inevitably suffers from a performance drop but still performs better than other state-of-the-art methods. As for Splatter Image [2], it also meets a significant performance drop when random input views are used as its SSIM $\uparrow$ decreased from 0.9151 to 0.8932 and LPIPS $\downarrow$ increased from 0.1517 to 0.2575 despite its PSNR $\uparrow$ has a slight increase. We visualize the results of the two settings to show the difference in Figure 14 in the revised PDF. Therefore, it does not mean Splatter Image [2] demonstrates superior generalization regarding input view pose distribution, but it appears that the PSNR $\uparrow$ of Splatter Image [2] does not increase when the setting is switched from random input view to fixed input view, which might be caused by some its inherent problems. We add the analysis in our Appendix in the revised version.
>
> [1] Xuanyu Yi, et al., MVGamba: Unify 3D Content Generation as State Space Sequence Modeling. arXiv2024.
> [2] Stanislaw Szymanowicz, et al., Splatter Image: Ultra-Fast Single-View 3D Reconstruction. CVPR2024.
> [3] Jiale Xu, et al., InstantMesh: Efficient 3D Mesh Generation from a Single Image with Sparse-view Large Reconstruction Mod-
> els. arXiv2404.
> [4] David Charatan, et al., pixelSplat: 3D Gaussian Splats from Image Pairs for Scalable Generalizable 3D Reconstruction. CVPR2024.
> [5] Jiaxiang Tang, et al., LGM: Large Multi-View Gaussian Model for High-Resolution 3D Content Creation. ECCV2024.

---

> ### Author Response · Authors · 2024-11-25
>
> Dear Reviewer 4FKX,
>
> We want to express our sincere gratitude for your insightful suggestions which are instrumental in enhancing the quality of our work. We hope that our proposed modifications would have addressed your concerns about the clarity of our presentation. We would really appreciate it if you could let us know if there are any further questions or aspects of the paper that require additional clarification.
> Thank you once again for your time and consideration.

---

> ### Author Response · Authors · 2024-11-27
>
> Dear Reviewer 4FKX,
>
> For weakness1, we now provide the updated visualization with resolution 512 in the revised PDF Appendix Figure 20. Should you have any further questions or concerns, please do not hesitate to reach out to us.

---

> > ### Author Response · Authors · 2024-12-02
> >
> > Dear Reviewer 4FKX,
> >
> > Thank you for dedicating your time and providing feedback on our work. We have tried our best to address the concerns you previously raised by providing additional explanations or conducting further experiments, and we have rectified writing issues.
> > We kindly seek your thoughtful reconsideration for a potential score increase, taking into account of the revisions made based on your invaluable feedback if you have no further concerns. Your time and insights are sincerely valued. If you have further questions, we are also pleased to answer you in the rest discussion period.

---

> > > ### Author Response · Authors · 2024-12-03
> > >
> > > Dear Reviewer 4FKX,
> > >
> > > Thank you for dedicating your time and providing feedback on our work. We have presented new experiments and more explanations to address your concerns or questions. As it approaches the end of the discussion period, we really want to know do you still have other concerns or questions so that we can put efforts in the last day to solve them. Your reply is very important for us, and we are looking forward to it.

---

### Author Response · Authors · 2024-11-20
**We update the revised PDF and add supplimentary**

Thank you very much for your thoughtful comments on our work. We greatly appreciate your feedback. We will address each of your concerns individually as outlined below. For those that require additional experiments, we will ensure to upload the results as soon as possible. Should you have any further questions or concerns, please do not hesitate to reach out to us. We also provide the revised PDF and some videos in the supplimentary.

---

> ### Comment · Area_Chair_c4RT · 2024-11-25
>
> Dear Reviewers,
>
> Many thanks for your reviews on submissions of ICLR 2025. Could you please read the authors' rebuttals and give your replies?
>
> Best wishes,
>
> AC

---

### Meta-Review · Area_Chair_c4RT · 2024-12-17

**Metareview:**

The paper presents a view-consistent 3D reconstruction and novel view synthesis model using 3D Gaussians representation from sparse images. The main task is to solve view inconsistency from multiple images.

It must be that directly merging 3D Gaussians through point concatenation is not good. The general way to make 3D reconstruction from multiple images is to transform different camera coordinate systems into one coordinate system and then to make optimizations. No people directly make concatenation of different 3D results by each separate image. Therefore, the motivation is not a good idea. The paper needs rewriting to give importance for the proposed method.

Moreover, in spite of no comparison with GS-LRM and GRM, the comparisons with PixelSplat and MVSplat are not reasonable. PixelSplat and MVSplat are both scene reconstruction methods, not object-level reconstruction methods. Therefore, comparing them with object-level reconstruction methods would be unfair, even if trained on object-level datasets. Thus, outperforming them on object-level datasets does not necessarily demonstrate the superiority of the method.

 Also, the method can only support a relatively small number of Gaussians. This may restrict its scalability.

**Additional Comments On Reviewer Discussion:**

Reviewer 4FKX raised visual result problem and unobvious improvements compared with previous methods. The authors made new visualization by new resolution of 512  and gave explanation on PSNR. Reviewer 4FKX had no responses.

Reviewer 9662 raised the importance of the MVDFA module and the two stages are not convincing, inconsistency motivation is not important, there lacks visual comparison. The authors gave the differences with  Triplane Gaussian and Instant3D.  For the inconsistency, they also used rigid transformations to unify different images. They also gave some visualization results. The reviewer understood the differences between this work and Triplane Gaussian and Instant3D. But, the reviewer still had a question on the inconsistency.

Reviewer t736 proposed multiple highly relevant prior references are not cited. In particular, it lacks the comparisons with GS-LRM and GRM. Furthermore, Reviewer t736 had questions about the paper positioning and result quality. The authors discussed and added some experiments that addressed some concerns of Reviewer t736.  Reviewer t736 thought this discussion remained at an empirical level, not a theoretical one. The authors agreed that they were not able to prove the existence of the issue and modified the paper.

In the final discussions, the reviewers and AC thought there are still the problems: the result quality is not groundbreaking to be state-of-the-art, the positioning and motivation need reinvented, it is unclear to scale to larger scenes.

---

### Decision · Program_Chairs · 2025-01-22

Reject